

# Effects of intensified freeze-thaw frequency on dynamics of winter

# nitrogen resources in temperate grasslands

Chaoxue Zhang[1,2], Na Li[3], Linna Ma[1,2,*]

[1] State Key Laboratory of Forage Breeding-by-Design and Utilization, Institute of
Botany, the Chinese Academy of Sciences, Beijing 100093, China
[2] Key Laboratory of Vegetation and Environmental Change, Institute of Botany, the
Chinese Academy of Sciences, Beijing 100093, China
[3] Ministry of Education Key Laboratory of Ecology and Resource Use of the
Mongolian Plateau, School of Ecology and Environment, Inner Mongolia University,
Hohhot 010021, China

**\*Corresponding author:**
Email: maln@ibcas.ac.cn; Tel: +86-10-62836564






## Abstract

In seasonal snow-covered temperate regions, winter serves as a crucial phase for
nitrogen (N) accumulation through persistent mineralization processes. Climate
warming has accelerated snowmelt and intensified freeze-thaw cycle frequency (FTC),
potentially altering the availability of winter N sources for plants. We simulated
intensified FTC regimes (increased 0, 6, and 12 cycles) in situ across two contrasting
temperate grasslands, employing dual-labeled isotopes ($^{15}NH_4^{15}NO_3$) to quantify
winter N dynamics. Our results showed that intensified FTC significantly enhanced
soil net ammonification rates and inorganic N levels in early spring, while net
nitrification rates remained stable. This suggests that frequent FTC may provide a
substantial N source for soil microorganisms and plant growth. Notably, soil microbial
biomass N increased despite microbial C limitation, indicating efficient microbial N
competition that restricted plant access to winter N sources. Intensified low-frequency
FTC did not affect plant $^{15}N$ acquisition, whereas high-frequency FTC significantly
reduced plant $^{15}N$ acquisition. Importantly, the impacts of FTC on plant $^{15}N$
acquisition varied among functional types. Dominant cold-tolerant species (perennial
bunch grasses and semi-shrubs) increased $^{15}N$ acquisition, likely due to earlier root
activity, while subordinate species (perennial rhizome grasses and forbs) exhibited
reduced acquisition. In conclusion, while intensified FTC did not lead to the loss of
winter N sources, it restructures N availability by favoring microbial retention and
creating competitive hierarchies among plants in temperate grasslands. The
high-frequency FTC-induced shifts in partitioning of winter N resources could



substantially influence grassland productivity and community structure, highlighting
the critical need to integrate winter climate change effects into temperate grassland
ecosystem models.
**Keywords:** freeze-thaw cycle; grassland; N isotope; N dynamic; plant N acquisition;
snowmelt; winter



## 1 Introduction

Approximately 50 % of terrestrial ecosystems in the Northern Hemisphere experience

seasonal snow cover and winter soil freezing (Sommerfeld et al., 1993; IPCC, 2021).

Remarkably, soil microorganisms maintain metabolic activity under snowpack and

contribute to nutrient mineralization throughout winter (Larsen et al., 2012; Zhang et

al., 2011). These winter processes, including soil nitrogen (N) mineralization and

microbial N immobilization, constitute a vital nutrient reservoir that supports plant

growth across alpine grasslands, temperate grasslands, and boreal forests (Alatalo et

al., 2014; Bilbrough et al., 2000; Collins et al., 2017; Edwards and Jefferies, 2010).

The springtime release of winter-accumulated N (mainly including $NH_4^+$, $NO_3^-$, and

dissolved organic N) through freeze-thaw cycles (FTC) synchronizes nutrient

availability with plant demand (Kaiser et al., 2011), particularly in N-limited

ecosystems where winter N contributions may determine growing season productivity

(Schmidt and Lipson, 2004).

Climate warming has emerged as one of the most important global environmental

challenges. Evidence shows that climate warming has primarily occurred during

winter, with the rate of winter warming exceeding the annual average over the past

few decades in China (Zong and Shi, 2020). This trend is expected to intensify, with

an anticipated increase in the frequency of extreme warming events (Kreyling et al.,

2011; IPCC, 2021). Winter warming might lead to an earlier onset and intensified

frequency of freeze-thaw cycles (FTC), potentially altering ecosystem N cycling





processes (Gao et al., 2018). Consequently, this could affect the availability of winter
N sources for plant growth. However, how intensified FTC affect winter N retention
remains poorly understood, particularly its subsequent impacts on plant N uptake and
ecosystem functioning.
Intensified FTC induces complex shifts in soil N dynamics by simultaneously
enhancing N mineralization while disrupting microbial immobilization and ecosystem
retention processes. Existing research have demonstrated that intensified FTC can
enhance soil N availability in cold regions (Dai et al., 2020; Nie et al., 2024; Teepe
and Ludwig, 2004; Wang et al., 2012; Yang et al., 2023). The physical disruption
caused by FTC promotes the N release from both soil organic matter and microbial
biomass via cell lysis (Koponen et al., 2006; Sawicka et al., 2010; Skogland et al.,
1988). However, this FTC-induced N pulse often occurs before plants resume active
uptake, leading to substantial N losses through leaching and gaseous emissions (Chen
et al., 2021; Elrys et al., 2021; Ji et al., 2024). while microbial mortality reduces
microbial N immobilization (Gao et al., 2018), the surviving microbial community
exhibits stimulated microbial activity that accelerates nutrient cycling (Fitzhugh et al.,
2001; Nie et al., 2024; Sharma et al., 2006; Wang et al., 2024). Notably, a
comprehensive meta-analysis by Song et al. (2017) indicated that FTC have no
significant effect on microbial biomass N (MBN) across various ecosystems,
including forest, shrubland, grassland/meadow, cropland, tundra and wetland
ecosystems, suggesting complex compensatory mechanisms in microbial N retention.



Frequent FTC significantly impact plant-soil N dynamics through multiple pathways.
Root damage caused by FTC directly impairs plant N acquisition capacity (Campbell
et al., 2014; Song et al., 2017), while simultaneously creating temporal mismatches in
N availability. Larsen et al. (2012) utilized $^{15}$N tracer reveals that soil microorganisms
initially dominate winter N immobilization following snowmelt, with plant functional
types exhibiting sequential N uptake patterns: evergreen dwarf shrubs being the first
to take up winter N, succeeded by deciduous dwarf shrubs and graminoids in late
spring in the alpine ecosystem. This study highlighted a temporal differentiation in the
resumption of N uptake among plant functional groups after winter. This temporal
niche partitioning is particularly pronounced in temperate regions, where shallower
snowpack and more frequent spring FTC create distinct competitive environments
compared to alpine systems. Studies in temperate grasslands have shown that
perennial bunch grasses present earlier N uptake than perennial rhizome grasses and
forbs (Ma et al., 2018, 2020), a phenological advantage that becomes more
pronounced under winter warming conditions (Turner and Henry, 2009). These
findings collectively highlight how FTC-mediated changes in belowground processes
interact with plant functional traits to govern winter N partitioning.

While previous studies have examined winter N cycling in high-altitude and
high-latitude regions experiencing rapid warming trends (Alatalo et al., 2014;
Bilbrough et al., 2000; Brooks et al., 1996; Edwards and Jefferies, 2010; Miller et al.,



2007), temperate grasslands remain understudied despite their distinct freeze-thaw
regimes. Critically, existing research has predominantly relied on laboratory
simulations employing artificial freezing regimes (DeLuca et al., 1992; Teepe and
Ludwig, 2004; Yang et al., 2023; Zhang et al., 2024), creating significant gaps
regarding the ecological impacts of natural in situ freeze-thaw cycles. Field-based
investigations are urgently needed to address two critical questions: (1) how FTC
frequency alters winter N retention dynamics, and (2) whether these changes create
legacy effects on subsequent growing season productivity and plant community
composition in temperate grasslands. This lack of field evidence limits our ability to
predict ecosystem responses to climate change.

Temperate grasslands cover nearly 40 % of China's terrestrial ecosystems (Bardgett et
al., 2021), and are particularly vulnerable to climate change due to their prolonged
near-freezing winter conditions. To understand how intensified FTC affect retention
of winter N resources in grasslands, we conducted an in situ $^{15}NH_4^{15}NO_3$ tracer
experiment across two temperate grassland types. We hypothesize that: (1) intensified
FTC would reduce retention of winter N resources through physical disruption of soil
aggregates enhancing N mobility, root damage impairing plant N uptake capacity, and
microbial cell lysis leading to N leaching and denitrification losses (Fitzhugh et al.,
2001; Koponen et al., 2006; Nie et al., 2024; Sawicka et al., 2010; Sharma et al., 2006;
Skogland et al., 1988); and (2) intensified FTC would lead to differential utilization of
winter N sources among plant species, mediated by interspecific variations in their




competitive abilities, root system architecture (particularly rooting depth and winter
root activity), and temporal niche partitioning in growth phenology (Bilbrough et al.,
2000; Hosokawa et al., 2017; Ma et al., 2018, 2020). Specifically, we expected that
deep-rooted species with early spring green-up would increase their utilization of
winter N sources, while other plants may experience reduced N utilization due to root
damage (Campbell et al., 2014; Song et al., 2017).

**2 Methods**
**2.1 Experimental site**
We conducted parallel experiments in two contrasting temperate grassland ecosystems:
a meadow steppe and a sandy steppe (Table 1; Fig. 1). The meadow steppe was
situated at the Hulunber Grassland Ecosystem Observation and Research Station in
northeastern China (49°19' N, 120°02' E, 628 m), while the sandy steppe was located
at the Ordos Sandy Grassland Ecology Research Station in northern China (39°29' N,
110°11' E, 1290 m).

Both sites have a continental climate. The mean annual precipitation is 420 mm and
310 mm, and the mean annual temperature is -2-1 °C and 6.5 °C in the meadow
steppe and the sandy steppe, respectively (http://data.cma.cn/). The non-growing
season for the meadow steppe extends from late September to late April of the
following year, with a spring freeze-thaw period occurring from late March to late
April. In contrast, the non-growing season for the sandy steppe lasts from



mid-October to late March, with the spring freeze-thaw period occurring from late
February to late March. In the meadow steppe, persistent snow cover reached 20-25
cm depth during late winter (January-February), providing consistent thermal
insulation. In contrast, the sandy steppe exhibited shallower and more variable
snowpack (typically 10 cm depth) due to higher wind redistribution and lower
moisture retention. Under natural conditions, the meadow steppe in this study
underwent a total of 19 freeze-thaw cycles, while the sandy steppe experienced 21
freeze-thaw cycles in early spring (http://nm.cma.gov.cn/).

The meadow steppe features high plant diversity and fertile soils, while the sandy
steppe exhibits lower diversity and nutrient-poor, coarse-textured soils (Table 1). This
strategic pairing allows for comprehensive assessment of FTC impacts across varying
resource availability and community structures, as evidenced by significant baseline
differences in N dynamics between sites. According to the Chinese Soil Classification
(GB/T 17296-2009), the predominant soil type in meadow grassland is loam soil, and
which in sandy grassland is sandy loam soil. The meadow steppe soil has higher C
and N content but slightly lower pH compared to the sandy steppe soil (Table 1). In
the meadow steppe, the dominant plant species include perennial bunch grasses like
*Stipa baicalensis Roshev*, perennial rhizome grasses such as *Leymus chinensis* (Trin.)
Tzvel., as well as perennial forbs including *Carex pediformis* C. A. Mey., which
together cover approximately 78% of the site. In the sandy steppe, the dominant plant
species are *Cleistogenes squarrosa* (Trin). Keng., *Klasea centauroides* (L). Cass., and



*Hedysarum monglicum* Turez, covering about 70 % of the area (Table 1). The
complementary strengths of these ecosystems enable robust predictions about
grassland responses to changing winter climate regimes.

**2.2 Experimental design**
In late October 2020, eighteen 3 m × 3 m plots were established at each site, with a
3-meter buffer between neighboring plots. The experiment employed a randomized
block design with three treatments and six replicates per site: (1) control (ambient
FTC), (2) intensified low-frequency FTC (LFTC; + 6 times), and (3) intensified
high-frequency FTC (HFTC; + 12 times). These treatments were designed to simulate
projected increases in winter FTC frequency under climate change scenarios.

The treatment levels were based on historical climate data showing approximately 20
natural FTCs typically occur during winter and early spring at both sites (Table 1;
https://data.cma.cn/). According to the definition of freeze-thaw cycling, a
freeze-thaw cycle is defined as the process in which soil temperature (0-10 cm) rises
above 0 ℃ and then subsequent drops below 0 ℃ (Yanai et al., 2007). Therefore, the
intensified FTC correspond to total increases in 30 % (+ 6 times) and 60 % (+ 12
times) in the frequency of FTC during winter and spring seasons, respectively.

Within each plot, we established a fixed 1 m × 1 m subplot for [15]N tracing. Building
upon established [15]N tracing approaches (Ma et al. 2020; Bilbrough et al. 2000), we



applied $^{15}NH_4{}^{15}NO_3$ solution prior to winter soil freezing. A solution containing 24 mg
$^{15}N\ L^{-1}$ of $^{15}NH_4{}^{15}NO_3$ was injected into 100 holes with a syringe guided by a grid
frame (1 m × 1 m), with each hole receiving 2 mL of the labeled solution. The total
application per subplot was 200 mL, which is equal to 120 mg $^{15}N\ m^{-2}$. The added $^{15}N$
was kept within the natural fluctuation range of inorganic N in the soil, approximately
7 %-10 % of background soil inorganic N levels. We injected water into control
treatments instead of the $^{15}N$ tracer, and no significant differences in plant/microbial N
concentrations when compared to the $^{15}N$ treatments. This indicates that the $^{15}N$
application did not disrupt natural N cycling processes (Ma et al., 2018).

Based on recent 5-year climatic records, our initial FTC treatments were scheduled
approximately 15 days prior to the natural spring FTC period (late winter). For the
freezing-thaw manipulation, a closed-top tent (3 m length × 3 m width × 2 m
height) was installed in each plot during each warming manipulation. The heating
tents were constructed with polyester fabric, featuring sealed tops and mesh-sided
windows to prevent excessive $CO_2$ accumulation while maintaining temperature
control. Within each tent, we used a propane AirHeater (Mr Heater, USA) to raise soil
temperature to 2-3 °C (0-15cm), maintaining this temperature continuously for 8 to
10 hours each time. Continuous temperature logging was performed using 2
temperature detectors per treatment positioned at two critical positions: (a) 5 cm
above soil surface (ambient microclimate) and (b) 10 cm soil depth, with data
recorded at half-hour intervals throughout the experiment. The temperature was then



allowed to drop to approximately -2 °C over a period of 4 hours to complete one
freeze-thaw cycle. Two intensified FTC regimes were implemented: (i)
high-frequency FTC (HFTC) with 12 additional cycles administered every 1-3 days,
and (ii) low-frequency FTC (LFTC) with 6 additional cycles every 3-6 days. During
the natural freeze-thaw period, all artificial FTC treatments were deliberately
conducted when daily mean temperatures remained below -2 °C to avoid interference
with natural cycles.

**2.3 Sampling and Processing**
Samplings were conducted after the freeze-thaw treatments and during the succeeding
growing season. In the meadow steppe, samplings were collected on the following
dates: 26 March 2021 (early spring); 4 May 2021 (late spring); 23 June 2021 (early
summer); 22 July 2021 (late summer); and 26 September 2021 (late autumn).
Similarly, in the sandy steppe, samplings were collected on 5 March 2021 (early
spring); 29 April 2021 (late spring); 21 June 2021 (early summer); 26 July 2021 (late
summer); and 15 October 2021 (late autumn).

For plant samplings, soil blocks (20 cm length × 20 cm width × 20 cm height)
containing dominant plant species were carefully excavated and sectioned. Plant roots
were washed with distilled water to remove surface $^{15}N$, then separated into
aboveground and belowground components. All plant materials were oven-dried at
65 °C for 48 hours. For soil samplings, we randomly excavated three labeled core at



20 cm depth soil (diameter is 3.5 cm) from each plot. We combined three soil core
into a composite sample, which was passed through a 2 mm sieve. Within 4 hours of
collection, the composite sample was separated into two portions: one for air-drying
and soil analysis, and the other stored at -20 °C for microbial analysis.

Soil temperature and moisture at a depth of 10 cm were measured automatically every
half hour using HOBO data loggers (H21-USB, Onset Inc., USA) throughout the
study period. Soil and plant (aboveground and belowground) dry samples were
pulverized using a ball mill. Subsequently, soil samples were sieved through a
100-mesh sieve and plant samples through an 80-mesh sieve. The sieved samples
were analyzed for C and N content using an elemental analyzer (Elementar Vario Max
CN, Germany). Soil net ammonification and nitrification rates were analyzed using
the method of polyvinyl chloride plastic (PVC) core (Raison et al., 1987). A pair of
PVC cores was vertically inserted to 20 cm depth soil layer in each plot to incubate
the soil without plant uptake. One core was collected as the initial (unincubated)
sample to determine the concentrations of $NH_4^+$-N and $NO_3^-$-N using a flow injection
autoanalyzer (Scalar SANplus segmented flow analyzer, Netherlands). The other core
was incubated in situ for two weeks within capped cores. After incubation, we
analyzed the $NH_4^+$-N and $NO_3^-$-N in these samples as well. Net ammonification and
nitrification rates were estimated based on the changes in $NH_4^+$-N and $NO_3^-$-N levels
between the incubated and initial values. Soil total dissolved organic N (DON) was
calculated as total N minus inorganic N (i.e., the sum of $NH_4^+$-N and $NO_3^-$-N ). Soil





total dissolved organic C (DOC) was calculated as total C minus inorganic C.

The microbial biomass C and N were assessed by the fumigation-extraction
method with a total organic C analyzer (TOC multiN/C 3100, Analytik Jena, Germany;
Vance et al., 1987). The method calculates microbial biomass C and N by determining
the contrast in extractable C or N levels between samples that have been fumigated
and those that have not. To prepare the soil extracts, fresh soil samples are moistened
to a water retention capacity of 60 %, followed by incubation in the dark at a
temperature of 25 °C for a week. After incubation, samples with a moisture content
equivalent to 25 g of dry weight were fumigated with chloroform ($CHCl_3$) for a
duration of 24 hours. The soil sample was extracted by agitating it with shaking 60ml
$K_2SO_4$ solution for 30min. After filtration, the extractable concentration of organic C
or N was determined by elemental analyzer (Elementar Analyzer, Vario MaxCN,
Germany). The conversion coefficient is 0.45. The $^{15}N$ contents in plant (2 mg) and
soil subsamples (20 mg) were analyzed with an elemental analyzer coupled with an
isotope-ratio mass spectrometer (IRMS, Thermo Finnigan MAT DELTAplus XP,
USA). Soil microbial $^{15}N$ was measured using alkaline persulfate oxidation, followed
by a modified diffusion method (Stark and Hart, 1996; Zhou et al., 2003). Soil
immobilized $^{15}N$ was then calculated by subtracting microbial $^{15}N$ from soil total $^{15}N$
(Ma et al., 2018).

Soil microbial community structure was determined using the phospholipid fatty acid





(PLFA) method (Bossio and Scow, 1998). Changes in PLFAs reflect the viable
biomass of fungi and bacteria, as well as microbial community structure in situ soils.
The fatty acids a13:0, i14:0, i15:0, i16:0, i17:0, a17:0, 16:1ω7c, 17:1ω8c, 18:1ω5c,
18:1ω9t, 17:0cy, and 19:0cy were chosen as representative of the bacterial group. The
fatty acids 16:1ω5c, 18:2ω6,9c, and 18:1ω9c were selected to represent the fungal
group.

**2.4 Statistical Analysis**
The $^{15}$N acquisition (% of applied $^{15}$N) in the shoot and root were calculated as: $[(^{15}N_I$
$- ^{15}N_a) \times$ biomass$/^{15}N_t] \times 100$, where $^{15}N_I$ and $^{15}N_a$ are the $^{15}$N concentrations
(g$^{15}$N g$^{-1}$ sample) in the labeled and the control samples; biomass is the shoot or root
biomass at each sampling time (g m$^{-2}$), and $^{15}N_t$ is the amount of total added $^{15}$N
tracer (g $^{15}$N m$^{-2}$).

The soil or microbial biomass $^{15}$N recovery (% of applied $^{15}$N) was calculated as:
$[(^{15}N_I - ^{15}N_a) \times V \times BD /^{15}N_t] \times 100$, where V represents the soil volume of the 20
cm soil profile (cm$^3$ m$^{-2}$), and BD is the bulk density (g cm$^{-3}$). Differences in soil,
microbial and plant properties, and $^{15}$N tracer retention in plants, soil, and
microorganisms between the two grasslands were analyzed using One-way ANOVA.
Repeated measurement ANOVA was used to analyze the influences of different FTC
treatments, sampling times, and grassland types on the measured indicators. Spearman
correlation coefficients between variables were calculated using "rcorr". To assess the



relative importance of predictors for plant ¹⁵N acquisition capacity, a random forest
model was constructed using the randomForest and rfPermute packages. Model
training utilized 70 % of the dataset for parameter optimization, with the remaining
30 % reserved for model validation. All above mentioned analysis were conducted
with SPSS 21.0 software (SPSS for Windows, Chicago, IL, USA) and RStudio
2025.05.0 (Posit Software, Boston, MA, USA), and graphics were plotted using
SigmaPlot, 14.0,Origin 14.0 and R Studio.

**3 Results**
**3.1 Soil microclimate**
The edaphic conditions, including soil total C content, inorganic N content, and
texture, exhibited significant differences between the two temperate grasslands ($p <$
0.05; Table 1). Throughout the winter freezing period, the lowest soil temperatures
(0−10cm) were about -23 °C in the meadow steppe and -20 °C in the sandy steppe,
respectively (Fig. 2a, b). In early spring, soil temperatures rose rapidly, accompanied
by significant snowmelt. Intensified low-frequency FTCs (LFTC) and high-frequency
FTCs (HFTC) enhanced soil moisture by 0.03 m³ m⁻³ and 0.05 m³ m⁻³, respectively,
and raised soil temperatures by 2 °C and 3 °C during the period. However, neither
intensified LFTC nor HFTC had any significant impact on soil moisture or
temperature in the subsequent growing season (Fig. 2a, b).

**3.2 Soil characteristics**





Throughout the study period, intensified LFTC and HFTC only significantly
increased soil $NH_4^+$-N levels in spring, but did not show any significant effects in the
following season ($p < 0.05$, Fig. 3a, b). In the meadow steppe, intensified LFTC and
HFTC significantly increased soil $NH_4^+$-N levels by 25.0 % and 24.0 % in late spring,
respectively ($p < 0.05$, Fig. 3a). Additionally, intensified LFTC enhanced net
ammonification rates by 44.3 % and 58.6 %, and HFTC increased them by 58.3 %
and 50.3 % in early and late spring, respectively ($p < 0.05$, Fig. 3e). In the sandy
steppe, LFTC and HFTC increased soil $NH_4^+$-N by 25.0 % and 23.3 % in late spring
($p < 0.05$, Fig. 3b). Intensified LFTC had no significant impact on net ammonification
rates, while HFTC increased net ammonification rates by 16.2 %, 63.3 %, and 37.2 %
in early spring, late spring, and early summer, respectively ($p < 0.05$, Fig. 3f). It is
important to note that neither LFTC nor HFTC had significant effects on $NO_3^-$-N or
net nitrification rates at either site throughout the study period ($p < 0.05$, Fig. 3c, d, g,
h).

Intensified LFTC significantly decreased the soil microbial biomass C (MBC) in
spring, while the effect of HFTC on microbial biomass N (MBN) persisted to summer
($p < 0.05$, Fig. 4a-d). In the meadow steppe, HFTC decreased MBC by 16.2 % ($p <$
0.05, Fig. 4a). Conversely, both LFTC and HFTC increased MBN by 26.2 % and
26.9 %, respectively ($p < 0.05$, Fig. 4c). In the sandy steppe, HFTC decreased MBC
by 11.3% in early spring. Unlike MBC, both LFTC and HFTC increased MBN by
8.5 % and 28.2 %, respectively ($p < 0.05$, Fig. 4b, d).




### 3.3 Plant properties


Intensified LFTC did not have significant influences on shoot and root biomass N of

the selected plant species during the growing season at either site (Fig. 5a-f). In the

meadow steppe, HFTC increased shoot and root biomass N of *Stipa baicalensis*

(perennial bunch grass) by 19.7 % and 21.8 % at the end of the growing season,

respectively. Conversely, HFTC decreased shoot and root biomass N of *Leymus*

*chinensis* (perennial rhizome grass) by 23.9 % and 16.2 %, and decreased those of

*Carex pediformis* (perennial forb) by 22.2 % and 18.0 % ($p < 0.05$, Fig. 5a, c, e). In

the sandy steppe, HFTC increased the shoot and root biomass N of *Hedysarum*

*monglicum* (semi-shrub) by 22.6 % and 23.7 %, respectively. However, HFTC

decreased those of *Cleistogenes squarrosa* (perennial bunch grass) by 25.3 % and

12.1 %, and those of *Klasea centauroides* (perennial forb) by 23.1 % and 20.3 %. ($p <$

0.05, Fig. 5b, d, f).

323

### 3.4 $^{15}$N Retention in the soil-microorganism-plant systems

In both grassland types, soil $^{15}$N recovery was highest in early spring, followed by a

rapid decline from late spring to late summer. This was then followed by a gradual

increase in recovery until late autumn (Fig. 6c, d). In contrast, plant $^{15}$N acquisition

increased steadily throughout the growing season in both grasslands, while microbial

$^{15}$N recovery exhibited only modest fluctuations over the entire growing season ($p <$

0.05, Fig. 6e-h).




During the early growing season, intensified LFTC had no significant effect on total
$^{15}$N recovery in soil-microorganism-plant systems, while intensified HFTC
significantly increased total $^{15}$N recovery (Fig. 6a, b). LFTC did not significantly
impact soil $^5$N recovery, but HFTC significantly increased soil $^{15}$N recovery in the two
grasslands ($p < 0.05$, Fig. 6c, d). In the meadow steppe, intensified LFTC and HFTC
significantly enhanced microbial $^{15}$N recovery by 38.0% and 26.6%, respectively, and
by 49.5 % and 32.5 % in the sandy steppe ($p < 0.05$, Fig. 6e, f). Intensified LFTC did
not significantly impact plant $^5$N acquisition; in contrast, intensified HFTC
significantly decreased plant $^{15}$N recovery in the two grasslands ($p < 0.05$, Fig. 6g, h).

In the meadow steppe, the $^{15}$N acquisition in the shoots of *Stipa baicalensis* (perennial
bunch grass) and *Carex pediformis* (perennial forb) were comparable, while *Leymus*
*chinensis* (perennial rhizome grass) exhibited lower $^{15}$N acquisition. In contrast, the
highest $^{15}$N acquisition in roots was observed in *Leymus chinensis*, followed by *Carex*
*pediformis* and *Stipa baicalensis*. In the sandy steppe, both shoot and root $^{15}$N
acquisition of *Hedysarum monglicum* (semi-shrub) were the highest among the
studied species. This was followed by the shoot $^{15}$N acquisition of *Cleistogenes*
*squarrosa* (perennial bunch grass) and *Klasea centauroides* (perennial forb). Notably,
the root $^{15}$N acquisition of *Klasea centauroides* was higher than that of *Cleistogenes*
*squarrosa*.



In the meadow steppe, HFTC increased shoot and root $^{15}$N acquisition of *Stipa*
*baicalensis* by 5.8 % and 9.3 %, respectively. In contrast, HFTC decreased $^{15}$N
acquisition in *Leymus chinensis* by 16.4 % and 12.1 %, and in *Carex pediformis* by
4.9 % and 7.8 % ($p < 0.05$, Fig. 7a, c, e). In the sandy steppe, shoot and root $^{15}$N
acquisition of *Hedysarum monglicum* increased by 3.8 % and 18.4 %, respectively.
Conversely, *Cleistogenes squarrosa* experienced decreases of 16.7 % in shoots and
14.4 % in roots, while *Klasea centauroides* showed the largest reductions, with
decreases of 16.1 % and 14.1 % in shoot and root $^{15}$N acquisition, respectively ($p <$
0.05, Fig. 7b, d, f).

**3.5 Controls on plant N acquisition**
In meadow steppe, correlation analysis revealed that plant $^{15}$N acquisition exhibited
the strongest positive correlation with soil temperature, followed by bacterial biomass,
soil $NO_3^-$-N levels, soil moisture, soil DOC and fungal biomass under LFTC ($p < 0.05$,
Fig. 8a). Negative correlations between net mineralization rates and plant $^{15}$N
acquisition were observed. Under HFTC, plant $^{15}$N acquisition showed the strongest
positive correlation with soil temperature, while bacterial biomass, soil $NO_3^-$-N levels,
soil dissolved organic C, microbial biomass C, and soil moisture also exhibited
significant positive correlations. In contrast, net nitrification rate, soil $NH_4^+$-N levels,
and soil total N content displayed significant negative correlations with plant $^{15}$N
acquisition ($p < 0.05$, Fig. 8b).



In sandy steppe, plant $^{15}$N acquisition exhibited the strongest positive correlation with
soil total C content, followed with microbial biomass N, soil total N content, soil
temperature, soil dissolved organic Cand N, and soil $NO_3^-$-N levels Under LFTC ($p <$
0.05, Fig. 8c). Conversely, fungal biomass, microbial biomass C, microbial biomass
C-to-N ratio, and bacterial biomass demonstrated the most significant negative
correlations with plant $^{15}$N acquisition ($p < 0.05$, Fig. 8c). Under HFTC, plant $^{15}$N
acquisition exhibited the strongest positive correlation with soil total N content, while
significant positive correlations were also observed with soil total C content, soil
temperature, microbial biomass N, soil dissolved organic C and N. In contrast, the
microbial C-to-N ratio, and microbial biomass showed significant negative
correlations with plant $^{15}$N acquisition ($p < 0.05$, Fig. 8d).

Random forest analysis further identified soil temperature as the primary predictor of
plants $^{15}$N acquisition capability across all treatments ($p < 0.05$, Fig. 9). In the
meadow steppe, dominant predictors shifted from soil total N, bacterial biomass, and
net nitrification rate under LFTC to a broader suite of predictors including soil total N,
bacterial biomass, soil moisture, soil $NH_4^+$-N levels, and soil dissolved organic C
content under HFTC ($p < 0.05$, Fig. 9a, b). In the sandy steppe, bacterial biomass,
fungal biomass, soil $NO_3^-$-N levels, and soil total N were key predictors under LFTC,
while soil moisture, soil total N, bacterial biomass, net nitrification rate, and net
ammonification rate were important predictors under HFTC ($p < 0.05$, Fig. 9c, d).





## 4 Discussion

### 4.1 Contrasting FTC sensitivity in temperate grasslands

This study investigated how intensified freeze-thaw cycles (FTC) affect dynamics of

winter N sources during subsequent growing seasons in two contrasting temperate

grasslands, using a $^{15}NH_4^{15}NO_3$ tracer. Contrary to our first hypothesis, intensified

FTC significantly enhanced soil net ammonification rates and inorganic N levels,

though net nitrification rates remained unaffected (Fig. 3). Our findings are consistent

with previous research indicating that intensified FTC significantly enhanced spring

soil inorganic N content across diverse ecosystems, including temperate forests,

alpine meadows, and wetlands (Dai et al., 2020; Ji et al., 2024; Nie et al., 2024; Teepe

and Ludwig, 2004; Wang et al., 2012; Yang et al., 2023). A possible explanation for

this phenomenon is that frequent FTC promote the release of DON through the

physical disruption of soil aggregates and microbial lysis (Koponen et al., 2006;

Sawicka et al., 2010; Skogland et al., 1988). This process likely stimulates microbial

activity, accelerating mineralization processes (Fitzhugh et al., 2001; Nie et al., 2024;

Sharma et al., 2006). The lack of significant change in $NO_3^-$ levels could be attributed

to leaching and minimal variations in nitrification rates (Gao et al., 2018). Notably,

substantial retention of soil N and soil microbial biomass N (MBN) (Figure 4, 6)

suggests that intensified FTC in early spring did not result in substantial loss of winter

N sources, but instead enhanced N availability for plants and soil microbial growth.

The meadow steppe showed greater sensitivity to FTC than the sandy steppe, with

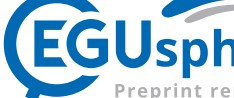

even increased low-frequency FTC enhancing net ammonification rates (Fig. 3).
Following spring thaw, the effects of FTC on ammonification rates and $^{15}$N retention
gradually diminished. suggesting these perturbations primarily create early-season
pulses rather than sustained changes. The differential response in the sandy steppe
could reflect its coarser texture, lower organic matter content, and more
drought-adapted microbial community that may be inherently more resistant to
FTC-induced disturbance (Yanai et al., 2007; Lipson et al., 2000). The contrasting
sensitivity between the two grasslands highlights the importance of considering
ecosystem-specific characteristics when predicting biogeochemical responses to
climate change.

**4.2 Intensified FTC alters microbial nutrient-use strategies**
Our study found that intensified FTC significantly reduced soil microbial biomass C
(MBC) during the early growing season in both grasslands (Fig. 4), consistent with
previous observations of microbial lysis under FTC conditions (DeLuca et al., 1992;
Walker et al., 2006). This reduction in MBC is mechanistically connected to DOC
loss due to intensified FTC (Deng et al., 2023). During snowmelt, soil DOC pool
becomes susceptible to leaching, particularly in early spring when plant uptake is
minimal, leading to a transient C limitation that further constrains MBC recovery
(Lipson et al., 2000; Sullivan et al., 2020). Interestingly, although significant
decreases in MBC were observed, microbial biomass N (MBN) presented significant
increases in early spring (Fig. 4). This increase in MBN may be attributed to enhanced



ammonification rates, which allows soil microorganisms to luxuriously utilize soil
inorganic N (Christopher et al., 2008; Nielsen et al., 2001; Skogland et al., 1988;
Wang et al., 2024; Yu et al., 2011). This decoupled response between MBC and MBN
suggests that soil microorganisms can effectively compete for winter-accumulated N
even when C becomes limiting, highlighting their adaptive capacity to prioritize N
storage under environmental stress (Yu et al., 2011). This also indicates that
FTC-induced stress triggers shift in microbial stoichiometry to optimize N retention at
the expense of C use efficiency (Schimel and Bennett, 2004), underscoring how
winter climate change may fundamentally alter microbial nutrient cycling strategies in
temperate grasslands.

**4.3 Limited losses of winter N resources under intensified FTC**
Our study reveals that intensified FTC did not cause significant losses of winter N
resources in temperate grassland ecosystems. We observed that intensified FTC
resulted in an increase in total $^{15}$N recovery within soil-microorganism-plant systems
during the early growing season, though this effect diminished over time and
eventually returned to ambient levels (Fig. 6 a, b), indicating that effective
ecosystem-level N retention mechanisms. The observed N retention capacity suggests
these ecosystems may be more resistant to winter climate change than previously
assumed (Han et al., 2018; Song et al., 2017). First, soil $^{15}$N recovery remained
significantly elevated throughout the entire growing season following FTC, indicating
efficient physical protection and chemical stabilization of released N within the soil



pool (Fig. 6). Second, microbial $^{15}$N recovery increased during the early growing
season after intensified FTC but gradually returned to ambient levels in the
mid-growing season (Fig. 6). This dynamic suggests that soil microorganisms rapidly
immobilized winter N resources during early spring, then progressively released it to
support subsequent plant growth (Bilbrough et al., 2000; Zheng et al., 2024; Turner
and Henry, 2009). Furthermore, the temporal decoupling of microbial N
immobilization and plant N uptake may serve as an important stabilizing mechanism
of winter N resources, preventing competitive exclusion while fostering mutually
beneficial plant-microbe interactions (Ma et al. 2020).

**4.4 Divergent plant strategies for $^{15}$N acquisition under intensified FTC**
Our findings partially support second hypothesis, revealing that high-frequency FTC
significantly reduced overall plant $^{15}$N acquisition, with contrasting responses
observed among different plant functional types (Fig. 7). While dominant species (*S.*
*baicalensis* in the meadow steppe and *H. monglicum* in the sandy steppe)
demonstrated enhanced $^{15}$N acquisition under intensified high-frequency FTC, other
perennial grasses (*L. chinensis* and *C. squarrosa*) and forbs (*C. pediformis* and *K.*
*centauroides*) showed reduced $^{15}$N uptake. These effects are likely attributed to
phenological differences in N acquisition timing, species-specific root system
vulnerability to FTC damage, and variation in competitive abilities under FTC stress
(Hosokawa et al., 2017; Reinmann et al., 2019; Song et al., 2017).





485 The increased $^{15}$N acquisition by dominant species likely reflects their ecological

486 adaptations to cold conditions (Fig. 5). *S.baicalensis*, as a cold-tolerant bunchgrass

487 with early spring phenology (Ma et al., 2018; Wang et al., 2016), and *H.monglicum*,

488 as a deep-rooted legume with nitrogen-fixing capability (Lonati et al., 2015), were

489 able to maintain root activity during freezing periods (Larsen et al., 2012) and

490 effectively compete for winter N sources. This aligns with observations that dominant

491 species can meet N demands during growing seasons through winter root activity

492 (Bilbrough et al., 2000; Miller et al., 2009).

493

494 The decreased $^{15}$N acquisition in subordinate species (perennial rhizome grasses and

495 forbs) reflects a complex physiological constraints and ecological trade-offs. Their

496 later N uptake phenology (Ma et al., 2018), coupled with greater susceptibility of fine

497 roots to FTC damage (Campbell et al., 2014; Song et al., 2017), limited their ability to

498 absorb winter N sources. First, the delayed N acquisition phenology of these species

499 creates a critical disadvantage. As demonstrated by Ma et al. (2018), perennial

500 rhizome grasses and forbs initiate $^{15}$N uptake significantly later than dominant bunch

501 grasses, particularly following freeze-thaw events, missing the early N pulse.

502 Additionally, intensified HFTC induces substantial fine root damage (Campbell et al.,

503 2014; Song et al., 2017), leading to increase mortality rates and constraining their

504 capacity to access winter N sources (Hosokawa et al., 2017; Reinmann et al., 2019).

505 In this study, the significant reduction in root biomass N supports these mechanistic

506 explanations (Fig. 5).




Second, although rhizome grasses have substantial coverage, they are less competitive
than bunch grasses and are more susceptible to environment disturbances, making
them vulnerable to damage during FTC period (Walker et al., 2004). Similarly,
perennial forbs (*Carex pediformis*) possess slender and creeping roots, which are also
prone to damage from FTC (Ye et al., 2017). These findings highlight intensified
HFTC may alter competitive hierarchies in grassland ecosystems by favoring
cold-adapted dominant species while disadvantaging other functional groups. Such
shifts could have important implications for plant community structure and ecosystem
functioning under changing winter climate conditions, particularly through potential
changes in N cycling dynamics and species composition.

**4.5 Future research directions**
To advance our understanding of intensified FTC effects on N dynamics in temperate
grasslands, future research should prioritize three key directions: first, molecular
characterization of cold-adapted microbial communities would elucidate the specific
taxa and functional genes responsible for retention of winter N resources during FTC
events, providing insights into the microbial mechanisms underpinning ecosystem
resilience; second, longer-term studies tracking $^{15}$N fate across multiple annual
freeze-thaw cycles are needed to assess whether the observed N retention patterns
persist over long timescales; finally, comparative investigations across diverse
grassland types and climatic gradients would help determine how soil properties,





vegetation composition, and regional climate modulate ecosystem responses to winter
climate change, enabling more accurate predictions of biogeochemical cycling under
future climate scenarios.

**5 Conclusions**
Our study provides novel mechanistic insights into how intensified frequency of
freeze-thaw cycles (FTC) regulate the dynamics of winter N sources in temperate
grasslands. Three important advanced emerge from our findings: first, the observed
decoupling of ammonification and nitrification processes under intensified FTC
reveals enhanced retention of winter N resources. Second, the microbial community
demonstrates remarkable adaptability to FTC stress, maintaining efficient N
immobilization despite carbon limitation, as evidenced by increased microbial
biomass N concurrent with decreased biomass C. Most importantly, intensified
high-frequency FTC reduced overall plant $^{15}$N acquisition, with divergent responses
among functional types: dominant cold-tolerant species (perennial bunch grasses and
semi-shrubs) maintained higher $^{15}$N acquisition through phenological advantages,
while subordinate species (perennial rhizome grasses and forbs) showed reduced
uptake. These findings indicate that microbial communities serve as resilient buffers
against N loss during FTC events, and plant functional traits mediate ecosystem-level
responses to changing winter conditions. The demonstrated partitioning patterns of
winter N resources challenge current models of grassland N cycling by revealing the
importance of winter processes in shaping growing season N availability. Future



research should focus on quantifying how these FTC-induced N dynamics scale to
influence multi-year ecosystem trajectories under climate change scenarios.

*Author Contributions*. L.M. and C.Z. conceived the project. C.Z. performed the field
experiments. C.Z. contributed datasets. C.Z. and N.L. interpreted the results. L.M. and
C.Z. wrote the manuscript.

*Acknowledgements*. The authors thank the Hulunber Grassland Ecosystem
Observation and Research Station, Chinese Academy of Agricultural Sciences and the
Ordos Sandy Grassland Ecology Research Station, Chinese Academy of Sciences for
help with logistics and access permission to the study site.

*Financial support*. The authors acknowledge the funding provided by the National
Natural Science Foundation of China (No. 32071602).

*Data availability statement*. All data are included in the manuscript.
*Conflict of interest*. The authors declare that they have no conflict of interest.

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




**Table 1** Climate, soil, and plant community properties (± Standard Error) in the meadow steppe
and the sandy steppe (n = 6).

| | Term | Meadow steppe | | Sandy steppe | |
|---|---|---|---|---|---|
| **Site information** | Location | 49°19' N, 120°02' E | | 39°29' N, 110°11' E | |
| | Soil type | Loam soil | | Sandy loam soil | |
| | MAT (℃) | -1.5~1 | | 6.5 | |
| | MAP (mm) | 420 | | 310 | |
| | Elevation (m) | 628 | | 1290 | |
| | Frequency of spring freeze-thaw cycle (times) | 19 | | 21 | |
| **Soil property** | STC (kg m$^{-2}$) | 3.98 ± 0.14* | | 1.00 ± 0.10 | |
| | SIN (g m$^{-2}$) | 1.79 ± 0.09* | | 0.86 ± 0.05 | |
| | 20-2000 μm (%) | 63.71 ± 1.58* | | 48.59 ± 1.98 | |
| | 2-20 μm (%) | 27.23.13 ± 0.63* | | 36.74 ± 067 | |
| | < 2 μm (%) | 10.13 ± 0.23* | | 6.42 ± 0.13 | |
| | pH | 7.36 ± 0.26 | | 8.57 ± 0.07 | |
| | BD | 1.37 ± 0.11 | | 1.26 ± 0.10 | |
| **Plant property** | Cover (%) | *Stipa baicalensis* | 40 ± 1.26 | *Hedysarum monglicum* | 35 ± 1.36 |
| | | *Leymus chinensis* | 20 ± 0.86 | *Cleistogenes squarrosa* | 23 ± 0.79 |
| | | *Carex pediformis* | 25 ± 0.59 | *Klasea centauroides* | 12 ± 0.48 |
| **The treatment times** | HFTC | 7 March, 9 March, 10 March, 12 March, 14 March, 15 March, 17 March, 18 March, 20 March, 21 March, 23 March, and 26 March 2021 | | 10 February, 16 February, 18 February, 20 February, 21 February, 23 February, 25 February, 26 February, 28 February, 1 March, 3 March, and 5 March 2021 | |
| | LFTC | 7 March, 10 March, 14 March, 17 March, 20 March, and 23 March 2021 | | 10 February, 18 February, 21 February, 25 February, 28 February, and 3 March 2021 | |

Significant differences between sites were identified using One-way ANOVAs: *, $p < 0.05$. MAT, mean annual temperature; MAP, mean annual precipitation; STC, soil total C content; SIN, soil inorganic N content; BD, soil bulk density; HFTC, increased high frequency freeze-thaw cycles (12 times); LFTC, increased low frequency freeze-thaw cycles (6 times).



**Figure Legends**

**Figure 1. Geographical distribution of the transect in a meadow steppe and a sandy steppe in northern China.**

**Figure 2. Soil temperature (Figure. 2a) and moisture (Figure. 2b) from autumn 2020 to autumn 2021 under intensified low-frequency freeze-thaw cycles (LFTC; 6 times) and high-frequency freeze-thaw cycles (HFTC; 12 times) treatments in a meadow steppe and a sandy steppe in northern China.** Shaded vertical bars indicate processing (treatment) period. Vertical lines indicate natural freeze-thaw periods. Nablas indicate sampling times, dates for [15]N tracer injection and sampling dates are also shown.

**Figure 3. Soil $NH_4^+$-N, $NO_3^-$-N, net ammoniation and net nitrification rates under intensified low-frequency freeze-thaw cycles (LFTC; 6 times) and high-frequency freeze-thaw cycles (HFTC; 12 times) treatments in a meadow steppe and a sandy steppe in northern China.** Vertical bars indicate the standard error (SE) of the means (n = 6). Different lowercase letters indicate statistically significant differences among treatment groups within sampling periods ($p <$ 0.05).

**Figure 4. Soil microbial biomass C and N under intensified low-frequency freeze-thaw cycles (LFTC; 6 times) and high-frequency freeze-thaw cycles (HFTC; 12 times) treatments in a meadow steppe and a sandy steppe in northern China.** Vertical bars indicate the standard error (SE) of the means (n = 6). Different lowercase letters indicate statistically significant differences among sampling periods ($p <$ 0.05).

**Figure 5. Plant biomass N (shoot and root) under intensified low-frequency freeze-thaw cycles (LFTC; 6 times) and high-frequency freeze-thaw cycles (HFTC; 12 times) treatments in a meadow steppe and a sandy steppe in northern China.** Vertical bars indicate the SE of the means (n = 6). Different lowercase letters indicate statistically significant differences among sampling periods (p < 0.05).

**Figure 6. Dynamics of [15]N tracers in soils, microorganisms, and plants under intensified**





low-frequency freeze-thaw cycles (LFTC; 6 times) and high-frequency freeze-thaw cycles
(HFTC; 12 times) treatments in a meadow steppe and a sandy steppe in northern China.
Vertical bars indicate the SE of the means (n = 6). Different lowercase letters indicate statistically
significant differences among sampling periods ($p < 0.05$).

Figure 7. Plant [15]N acquisition under intensified low-frequency freeze-thaw cycles (LFTC; 6
times) and high-frequency freeze-thaw cycles (HFTC; 12 times) treatments in a meadow
steppe and a sandy steppe. Vertical bars indicate the SE of the mean (n = 6). Different lowercase
letters indicate statistically significant differences among sampling periods *($p < 0.05$)*.

Figure 8. Relationships between plant [15]N acquisition and environmental predictors under
intensified low freeze-thaw cycle (LFTC; 6 times) and high freeze-thaw cycle (HFTC; 12
times) treatments in the meadow steppe and the sandy steppe. DOC represents dissolved
organic C content, DON represents dissolved organic N content, F:B denotes the fungal to
bacterial biomass ratio, and MBC:MBN indicates the microbial biomass C to N ratio.

Figure 9. Relative importance of environmental predictors for plant [15]N acquisition as
determined by random forest analysis under intensified low freeze-thaw cycle (LFTC; 6
times) and high freeze-thaw cycle (HFTC; 12 times) treatments in the meadow steppe and
the sandy steppe. Predictor importance is expressed as percentage increase in mean squared error
(%IncMSE) when each variable is permuted. DOC represents dissolved organic C content, DON
represents dissolved organic N content, F:B denotes the fungal to bacterial biomass ratio, and
MBC:MBN indicates the microbial biomass C to N ratio.



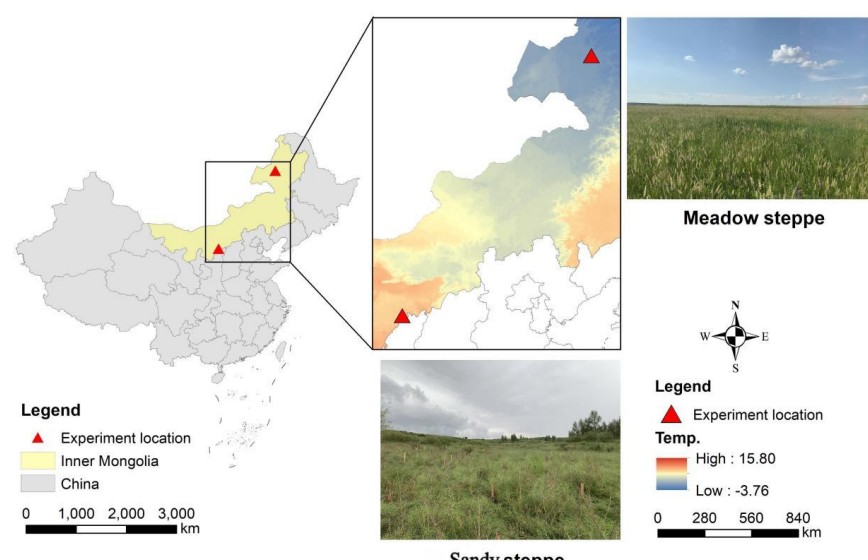

**Figure 1. Geographical distribution of the transect in a meadow steppe and a sandy steppe in northern China.**





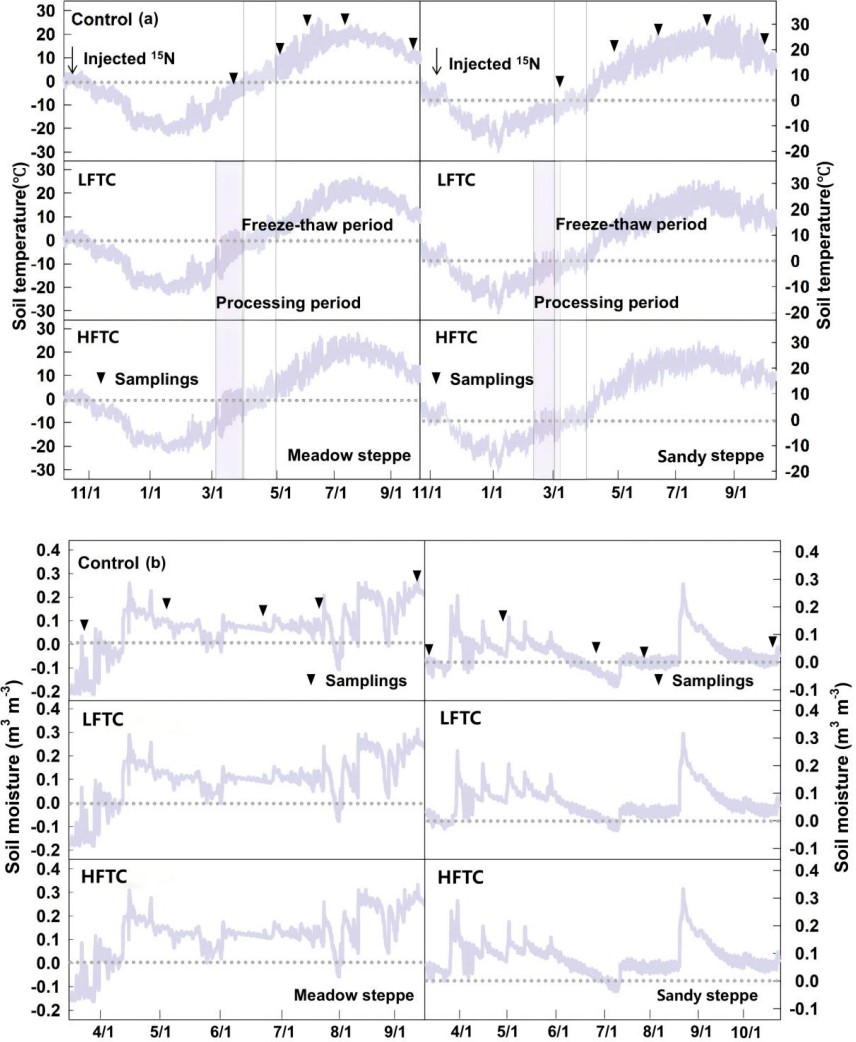

**Figure 2. Soil temperature (Figure. 2a) and moisture (Figure. 2b) from autumn 2020 to autumn 2021 under intensified low-frequency freeze-thaw cycles (LFTC; 6 times) and high-frequency freeze-thaw cycles (HFTC; 12 times) treatments in a meadow steppe and a sandy steppe in northern China.** Shaded vertical bars indicate processing (treatment) period. Vertical lines indicate natural freeze-thaw periods. Nablas indicate sampling times, dates for $^{15}$N tracer injection and sampling dates are also shown.



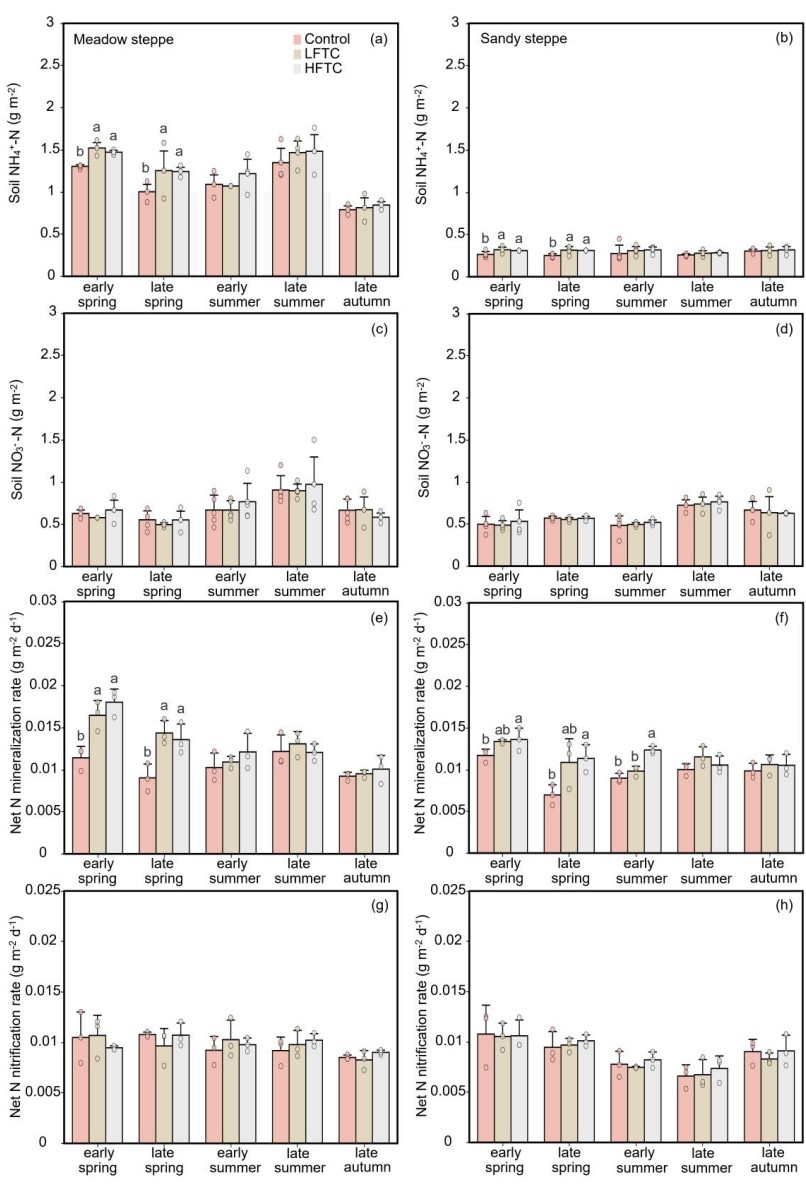

**Figure 3. Soil NH$_4^+$-N, NO$_3^-$-N, net ammoniation and net nitrification rates under intensified low-frequency freeze-thaw cycles (LFTC; 6 times) and high-frequency freeze-thaw cycles (HFTC; 12 times) treatments in a meadow steppe and a sandy steppe in northern China.** Vertical bars indicate the standard error (SE) of the means (n = 6). Different lowercase letters indicate statistically significant differences among treatment groups within sampling periods ($p < 0.05$).



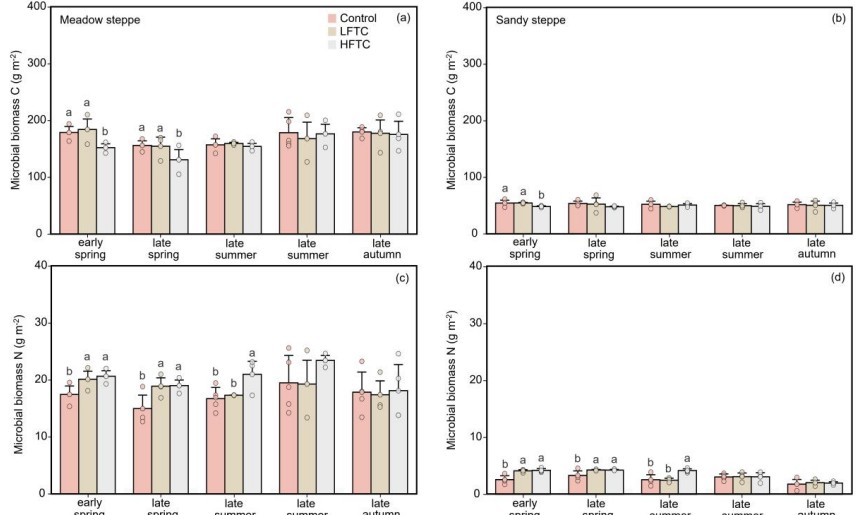

**Figure 4. Soil microbial biomass C and N under intensified low-frequency freeze-thaw cycles (LFTC; 6 times) and high-frequency freeze-thaw cycles (HFTC; 12 times) treatments in a meadow steppe and a sandy steppe in northern China.** Vertical bars indicate the standard error (SE) of the means (n = 6). Different lowercase letters indicate statistically significant differences among sampling periods ($p < 0.05$).



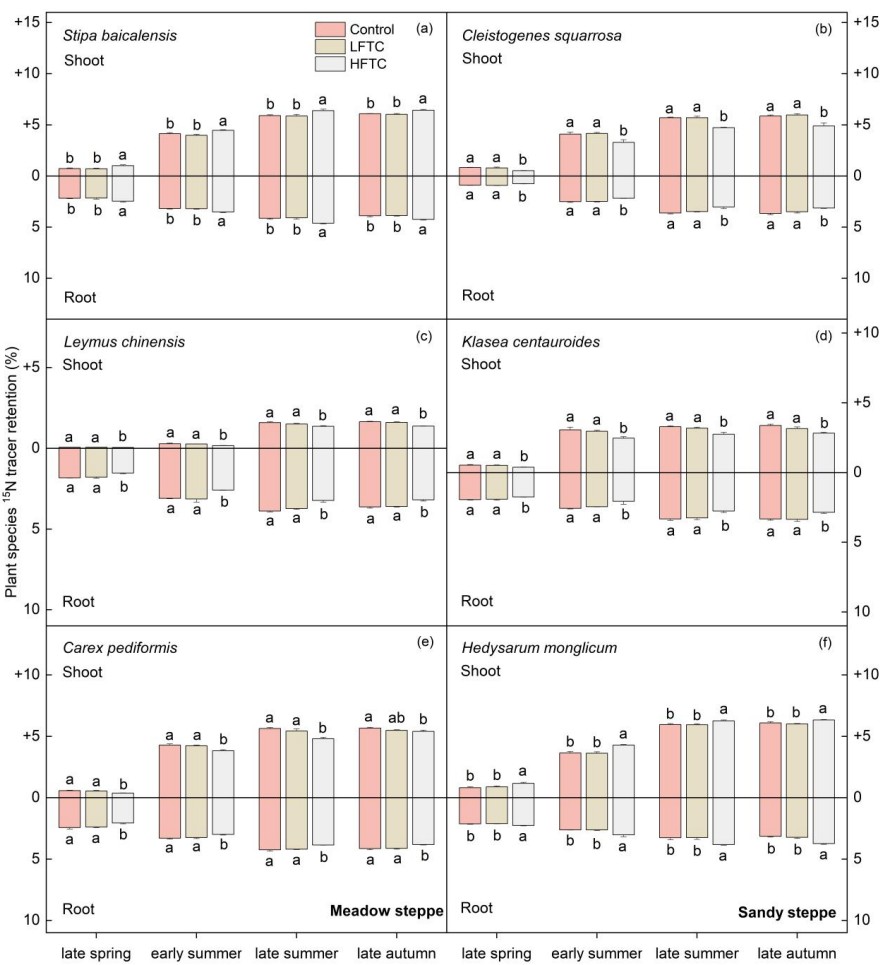

**Figure 5. Plant biomass N (shoot and root) under intensified low-frequency freeze-thaw cycles (LFTC; 6 times) and high-frequency freeze-thaw cycles (HFTC; 12 times) treatments in a meadow steppe and a sandy steppe in northern China.** Vertical bars indicate the SE of the means (n = 6). Different lowercase letters indicate statistically significant differences among sampling periods ($p < 0.05$).





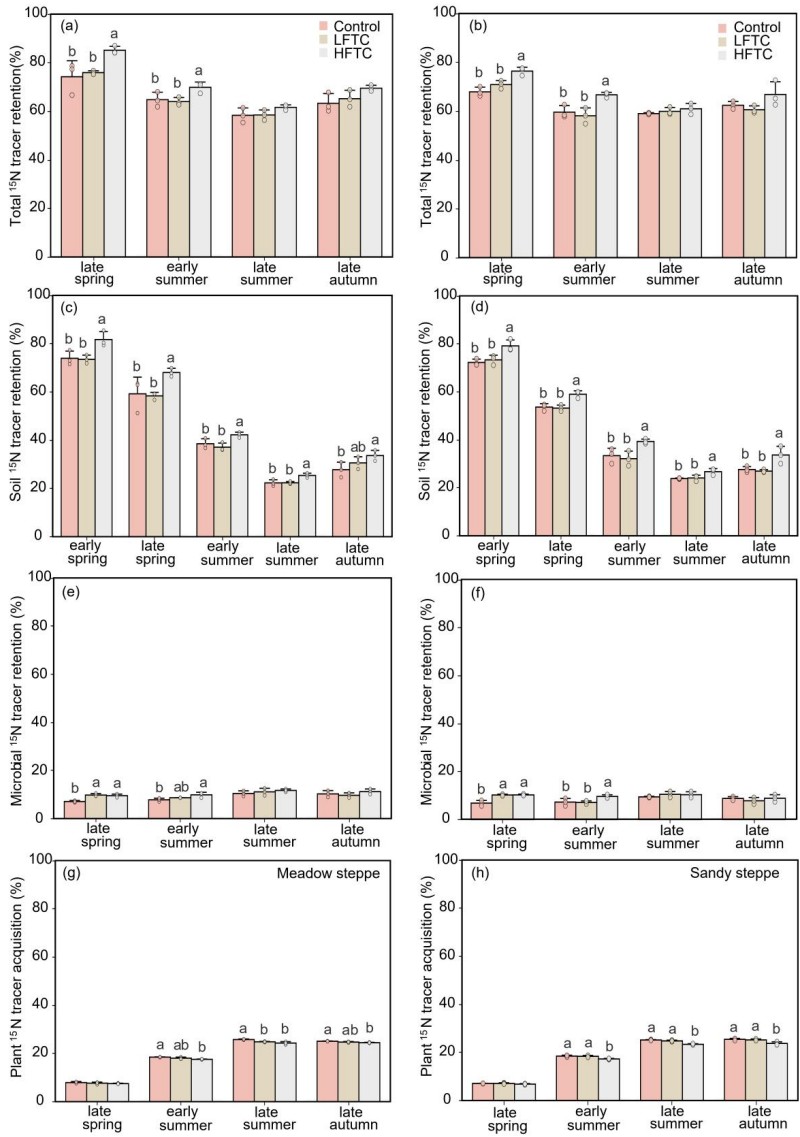

**Figure 6. Dynamics of $^{15}$N tracers in soils, microorganisms, and plants under intensified low-frequency freeze-thaw cycles (LFTC; 6 times) and high-frequency freeze-thaw cycles (HFTC; 12 times) treatments in a meadow steppe and a sandy steppe in northern China.** Vertical bars indicate the SE of the means (n = 6). Different lowercase letters indicate statistically significant differences among sampling periods ($p < 0.05$).



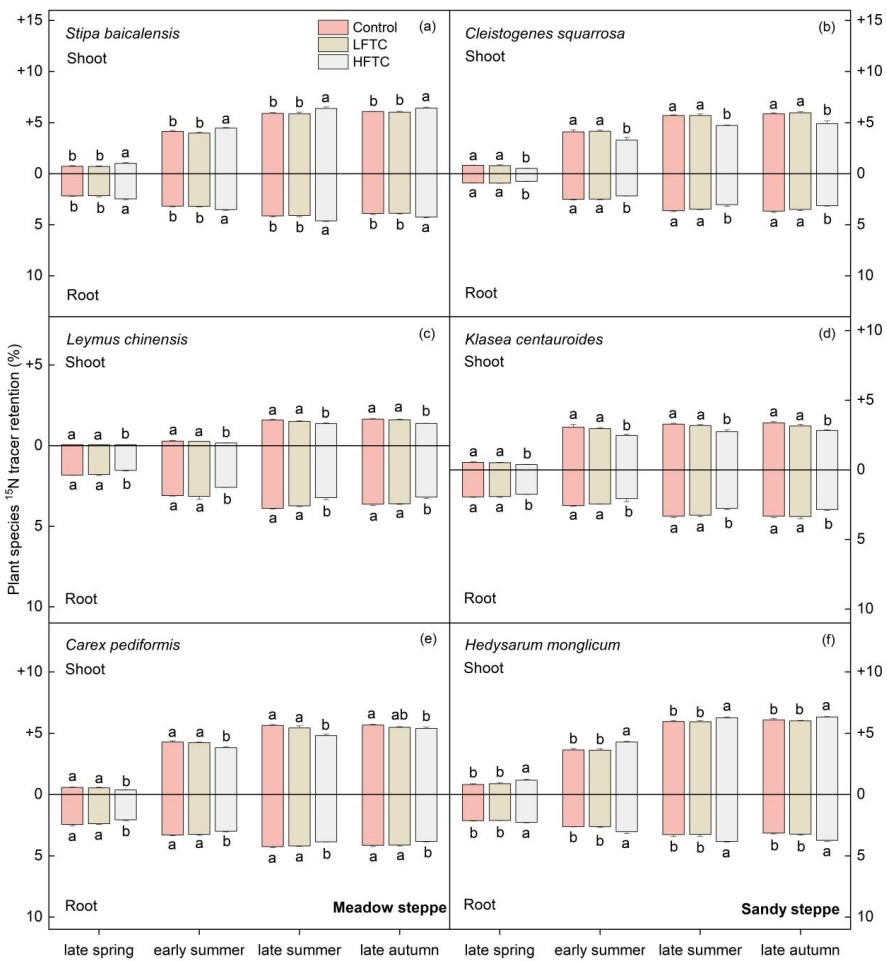

**Figure 7. Plant $^{15}$N acquisition under intensified low-frequency freeze-thaw cycles (LFTC; 6 times) and high-frequency freeze-thaw cycles (HFTC; 12 times) treatments in a meadow steppe and a sandy steppe.** Vertical bars indicate the SE of the mean (n = 6). Different lowercase letters indicate statistically significant differences among sampling periods ($p < 0.05$).



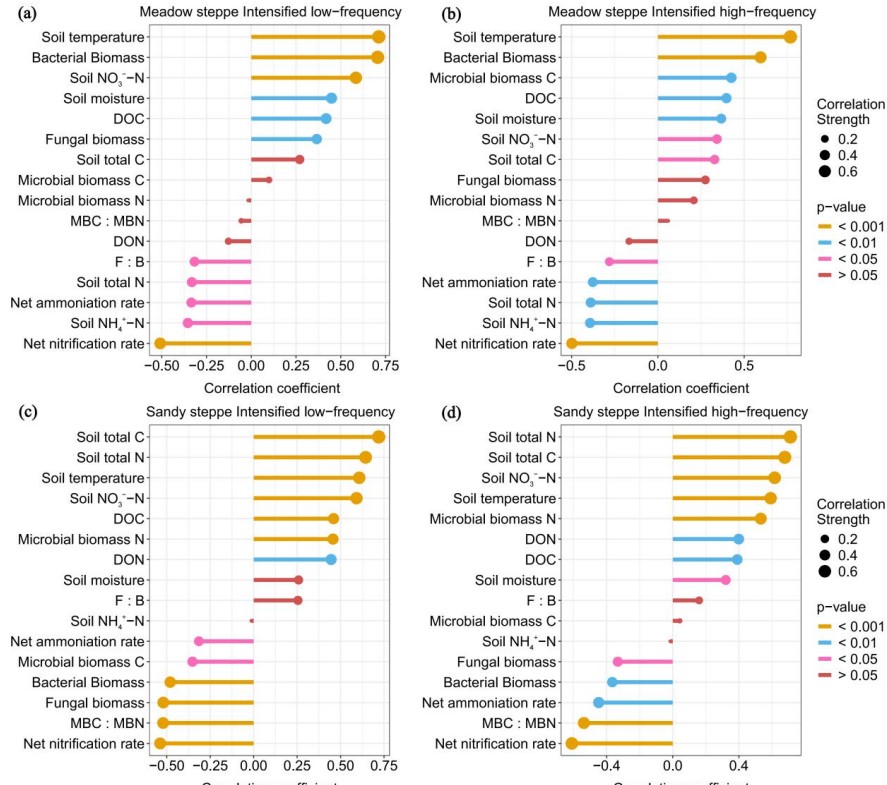

**Figure 8. Relationships between plant $^{15}$N acquisition and environmental predictors under intensified low freeze-thaw cycle (LFTC; 6 times) and high freeze-thaw cycle (HFTC; 12 times) treatments in the meadow steppe and the sandy steppe.** DOC represents dissolved organic C content, DON represents dissolved organic N content, F:B denotes the fungal to bacterial biomass ratio, and MBC:MBN indicates the microbial biomass C to N ratio.



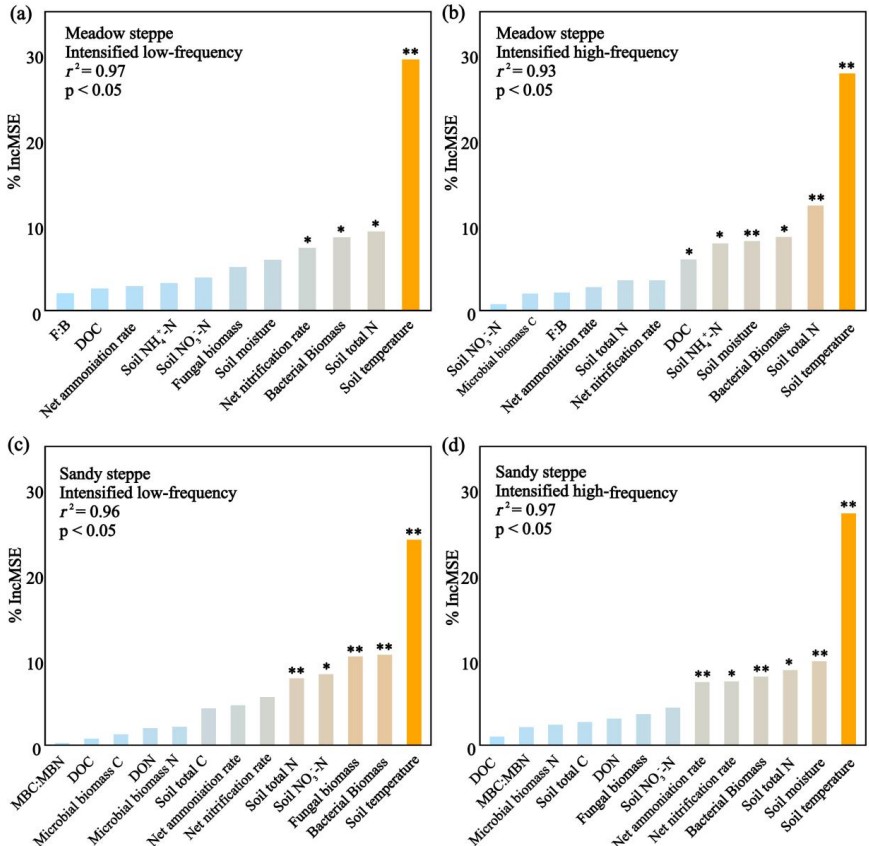

**Figure 9. Relative importance of environmental predictors for plant [15]N acquisition as determined by random forest analysis under intensified low freeze-thaw cycle (LFTC; 6 times) and high freeze-thaw cycle (HFTC; 12 times) treatments in the meadow steppe and the sandy steppe.** Predictor importance is expressed as percentage increase in mean squared error (%IncMSE) when each variable is permuted. DOC represents dissolved organic C content, DON represents dissolved organic N content, F:B denotes the fungal to bacterial biomass ratio, and MBC:MBN indicates the microbial biomass C to N ratio.