# Peer review of "Effects of intensified freeze-thaw frequency on dynamics of winter"

_EGUsphere, 2025_

## Author Comment (AC1)

**Manuscript Number: egusphere-2025-3080**

Title: Effects of intensified freeze-thaw frequency on dynamics of winter nitrogen resources in temperate grasslands

**Biogeosciences**

**Responses to reviewer #1:**

**Reviewer 1**

This manuscript presents a well-done designed and highly relevant study that investigates the effects of intensified freeze-thaw cycles (FTC) on winter nitrogen dynamics in temperate grasslands. The application of an in situ 15N tracer approach across two contrasting grassland sites represents a significant methodological strength, providing direct insights into the fate of winter N sources. The central finding—that intensified FTC restructures winter N availability by enhancing microbial retention and altering plant competitive hierarchies, rather than causing simple N losses—is novel, compelling, and has important implications for predicting ecosystem responses to winter climate change. The study is timely, addresses a critical knowledge gap, and possesses innovation.

While the scientific foundation of the work is strong and the conclusions are broadly supported by the data, the manuscript in its current form requires significant revision to fully realize its potential. I am enthusiastic about the potential of this manuscript to make a valuable contribution to the field. The necessary revisions are needed and primarily focused on presentation and interpretation. I am confident that after a thorough revision addressing the points above and those detailed in the specific comments, this manuscript will be suitable for publication in BG journal.

**Dear Editor and Reviewer:**

We sincerely thank the reviewer for the thoughtful and constructive comments on our manuscript. Each suggestion has significantly enhanced the depth and clarity of our study. We have carefully addressed every point raised and made comprehensive revisions throughout the manuscript to incorporate the reviewer's valuable feedback. These suggestion have strengthened the overall quality and impact of our work. Thank you once again for your time and expertise in helping us refine this research.

Best, Linna Ma

**Major concerns:**

(1) Structural and Narrative Flow: The organization of the Results and Discussion sections could be optimized to create a more logical. Specifically, the order of presenting findings could be rearranged to better guide the reader from the ecosystem-level outcome (N retention) down to the underlying mechanisms

**(soil processes, microbial uptake, plant competition).**

We thank the reviewer for this excellent suggestion regarding the manuscript's narrative flow. We agree that reorganizing the presentation to follow a clear hierarchy from ecosystem-level outcomes down to underlying mechanisms significantly improves the logical progression and reader comprehension.

We have restructured both the Results and Discussion sections to create a more coherent narrative. The revised order now systematically guides the reader as follows:

- (1) **Ecosystem-Level Outcome**: We first present the overarching finding regarding the fate of the winter-applied 15N, specifically, the total 15N recovery and retention at the ecosystem level, demonstrating that significant losses did not occur.
- (2) Underlying Mechanisms (Soil & Microbes): We then delve into the mechanisms responsible for this retention, first detailing the soil processes (e.g., ammonification, physical stabilization) and then the microbial response (e.g., shifts in biomass C and N, immobilization).
- (3) **Plant-Level Consequences**: Finally, we present the consequences for the plant community, showing how the altered N availability restructured competitive hierarchies and species-specific 15N acquisition.

We are confident that this reorganization has substantially improved the manuscript's clarity and impact, and we are grateful to the reviewer for this valuable recommendation.

**(2) The Results section currently contains lengthy lists of percentage changes.**

Thank you for your comment. We completely agree that this presentation style does not help to highlight the key findings and impairs the readability of the manuscript. We have rewritten the text in the Results section to be more narrative, guiding the reader to understand the meaning of the data rather than just the data itself. In the main text, we no longer list all data points one by one but instead focus on describing and emphasizing the most important and statistically significant trends and comparisons. We have retained only the most critical percentage values to support our core arguments.

(3) While generally clear, the manuscript requires a thorough proofread for grammatical consistency, conciseness, and precise scientific terminology.

We have undertaken a thorough revision of the entire manuscript with a specific focus on these aspects. This process included: comprehensive proofreading to correct grammatical errors and ensure consistency in tense, voice, and style throughout the text; a concerted effort to enhance conciseness by streamlining verbose sentences, removing redundant phrases, and improving the overall flow of the narrative.

(4) Discussion: The Discussion would benefit from stronger integration between sections (e.g., explicitly linking microbial decoupling to plant responses) and a more focused interpretation of the statistical analyses (e.g., highlighting the key drivers from the random forest analysis, rather than listing correlations).

We agree that strengthening the mechanistic narrative and providing a more focused interpretation of the statistical analyses are crucial for elevating the impact of our Discussion.

We have restructured the Discussion to create a logical causal chain and strengthen integration between sections and explicitly linking microbial processes to plant responses. In the newly structured section 4.2, we have proposed a clear, causal narrative that integrates the fate of nitrogen across the ecosystem components. From ecosystem outcome to underlying mechanisms, the section now begins by establishing the key ecosystem-level finding: the enhanced but transient total 15N recovery under HFTC. We then explicitly introduce the two primary retention mechanisms responsible for this pattern: the soil pool as a physical sink and microbial biomass as a biological buffer. We have directly addressed the link between microbial immobilization and subsequent plant availability as suggested, providing a more nuanced mechanistic understanding.

Regarding the random forest analysis, we have completely rewritten the relevant section (in 3.5) to move beyond a mere listing of correlations. We now clearly highlight and interpret the dominant predictors identified by the model. We discussed the potential biological and ecological reasons why these specific factors are the key drivers of the observed patterns. This provides a much clearer and more insightful interpretation of our statistical results.

**Specific comments:**

- 1. Abstract Line 34: Modify to read "employing the dual-labeled isotope". We don't see this on line 34, and we guess maybe line 36 needs to be changed. This has been revised as suggested.
- 2. Introduction Line 37: Change the period (.) before "while" to a comma (,). Revised as suggested.
- 3. Introduction Line 57: Modify to read "FTC results in distinct competitive environments".

Revised as suggested.

4. Line 58: Change "present" to "exhibit". Revised as suggested.

5. Line 59: Change "," to "-". Revised as suggested.

6. Line 82: Modify to read "(1) intensified FTCs reduce winter N retention through three primary mechanisms: (a) physical disruption of soil aggregates that enhances N mobility, (b) root damage that impairs plant N uptake capacity, and (c) microbial cell lysis that leads to N leaching and denitrification losses;".

Revised as suggested.

7. Line 82: Modify to read "intensified FTC would cause differential utilization of winter N sources among plant species. This effect is mediated by interspecific differences in three key traits: competitive ability, root system architecture (particularly rooting depth and winter root activity), and growth phenology (temporal niche partitioning)."

Revised as suggested.

8. Line 112: Modify to read "During the study period, the meadow steppe had a persistent snow cover that reached a depth of 20-25 cm in late winter".

Revised as suggested.

9. Line 114: Replace "exhibited" by "had".

Revised as suggested.

10. Line 117: Replace "underwent" by "experienced".

Revised as suggested.

11. Line 121: Modify to read "This contrast enables a comprehensive assessment of...".

Revised as suggested.

12. Line 125: Modify to read "...the predominant soil type is loam in the meadow steppe and sandy loam in the sandy steppe...".

Revised as suggested.

- 13. Line 132: Replace "(Trin)." by "(Trin.)" and replace "(L)." by "(L.)". Revised as suggested.
- 14. Line 133: It seems that there is extra space between the figure (70) and the unit (%).

Revised as suggested.

15. Line 155: Is it wrong here? If the amount of solution added is 120 mg 15N m-2, then it should be 600 mg 15N L-1.

Thanks for pointing this mistake. We have replaced "120" by "600".

16. Line 226: Replace "contrast" by "difference".

Revised as suggested.

17. Line 227: Replace "are" by "were".

The words "are" used instead of "were".

18. Line 231: Supplement the molar concentration of the K2SO4 solution, modify to read "...by shaking with 60 mL of a figure M K2SO4...".

We have corrected these words.

19. Line 233: Replace "elemental analyzer" by "total organic carbon analyzer".

Thanks. We have corrected these words.

20. Line 234: The conversion coefficient is abrupt, and it is integrated with line 225 to change: Microbial biomass C and N were calculated by dividing the differences in extractable C and N between fumigated and non-fumigated samples by a conversion factor of 0.45.

We have corrected these words.

21. Line 264: R package "rcorr" should be combined with version information. It is recommended to read: "Spearman correlation coefficients between variables were calculated using the rcorr function (in the Hmisc R package).".

Thanks. We have corrected these words.

22. Line 268: Replace "above mentioned" by "above-mentioned" and replace "analysis" by "analyses".

We have corrected these words.

23. Line 271: Replace "SigmaPlot, 14.0" by "SigmaPlot 14.0" and replace "R Studio" by "RStudio".

We have replaced "SigmaPlot, 14.0" by "SigmaPlot 14.0" and replaced "R Studio" by "RStudio".

24. Line 277: Replace "lowest" by "minimum".

The words "lowest" used instead of "minimum".

25. Line 297: Modify to read "In contrast, neither LFTC nor HFTC significantly affected NO3-N concentrations or net nitrification rates at either site during the study...".

We have corrected these words.

26. Line 302: Replace "LFTC" by "HFTC".

The words "LFTC" used instead of "HFTC".

27. Line 311: Modify to read "In contrast to the significant effects of HFTC, intensified LFTC had no significant impact on the shoot or root biomass N of the selected plant species at either site (Fig. 5a-f).".

We have corrected these words.

28. Line 315: The result section can be more concise and incorporate duplicate structures. Modify to read "HFTC significantly reduced biomass N in the perennial rhizome grass...(shoot: ...; root: ....)..." The sandy grassland is also changed in this way.

We have corrected these words.

29. Line 325: Replace "was highest" by "peaked".

The words "peaked" was used in the MS.

30. Line 338: Modify to read "In contrast to the positive effects on microbial recovery, HFTC significantly reduced plant 15N acquisition in both grasslands. LFTC had no significant effect on plant 15N recovery ...".

Revised as suggested.

31. Line 353: This paragraph also simplifies the language.

We have corrected these words.

32. Line 363: This section can be more concise and incorporate duplicate structures.

We have corrected these words.

33. Line 364: Modify to read "In the meadow steppe".

The word "the" was added in this sentence.

34. Line 367: Is it net nitrification rate or net nitrification rates? The whole article should be unified.

We have corrected these words. We also checked this across the whole MS.

35. Line 369: Modify to read "plant 15N acquisition showed the strongest positive correlation with soil temperature, while bacterial biomass, MBC, soil DOC, soil moisture, soil NO3-N levels, and soil total C also exhibited significant positive correlations.".

We have corrected these words.

36. Line 370: Replace "dissolved organic C" by "DOC" and replace

"microbial biomass C" by "MBC".

Revised as suggested.

37. Line 372: Modify to read "...soil total N content and net ammonification rate displayed...".

Revised as suggested.

38. Line 375: Modify to read "In the sandy steppe".

The word "the" was added in this sentence.

39. Line 375: According to correlation, rank the indicators from largest to smallest and long indicators are replaced with abbreviations.

Revised as suggested.

- 40. Line 377: It seems that a space is lacking between the "C" and the "and". Revised as suggested.
- 41. Line 418: Replace "with" by "as".

Revised as suggested.

42. Line 421: Replace "."by ",".

Revised as suggested.

43. Line 440: Replace "luxuriously utilize" by "engage in the luxury consumption of".

Revised as suggested.

44. Line 446: Replace "triggers shift" by "triggers a shift".

Revised as suggested.

---

## Author Comment (AC2)

**Manuscript Number: egusphere-2025-3080**

Title: Effects of intensified freeze-thaw frequency on dynamics of winter nitrogen resources in temperate grasslands

**Biogeosciences**

**Responses to reviewers #2:**

**General Comments**

This manuscript addresses an important topic in ecosystem nitrogen (N) cycling under changing winter conditions, using a 15N tracer approach to examine the fate of added nitrogen in plant, microbial, and soil pools following intensified freeze-thaw cycles (FTCs). The experimental design and tracer methodology are robust, and the study offers valuable insights into seasonal N allocation across ecosystem components.

However, several broader ecological interpretations—particularly those concerning ecosystem-level N retention, plant-microbial interactions, and species trait-based responses—go beyond what is directly supported by the data. Key processes such as N leaching, gaseous emissions, and root damage are not measured, limiting the conclusions that can be drawn about system-level outcomes and mechanisms.

Additionally, there are some inconsistencies between the hypotheses, measurements, and interpretations that should be addressed. Overall, the manuscript would benefit from a clearer distinction between the results directly observed in the data and the mechanisms proposed to explain them. The interpretation of the findings should be more cautious, and the methodological limitations of the study should be more explicitly acknowledged. Major revisions are needed to ensure that the hypotheses and conclusions accurately reflect the data obtained from the experiment, as some interpretations currently extend beyond what the results support. With these changes, the study could make a valuable contribution to understanding nitrogen cycling in seasonally frozen grassland systems.

**Dear reviewers:**

We sincerely thank you for your constructive comments on our manuscript. We have carefully addressed this point and made comprehensive revisions throughout the manuscript. Your suggestions have strengthened the overall quality of our work. Thank you once again for your time and expertise in helping us refine this research.

**The key improvements are summarized below:**

(1) Refined the hypotheses and Conclusions: We have rewritten our hypotheses to focus specifically on processes measurable with our 15N tracer data (i.e., contrasting plant community-level N retention, species-specific 15N acquisition), removing speculative mechanisms that were not directly measured (e.g., root damage, aggregate disruption). Correspondingly, all conclusions have been carefully revised to ensure they are fully supported by our data.

- (2) Incorporation of new data and analysis: In direct response to specific comments, we have incorporated new data on 15N leaching losses, which provides independent support for our conclusion of limited hydrological N loss. The interpretation of statistical analyses (correlation and random forest) has been refined to focus on the ecological meaning of key drivers rather than simply listing correlations.
- (3) Explicit methodological limitations: A new dedicated section ("Limitations and future work") has been added to the Discussion. This section explicitly acknowledges the scope of the 15N tracer method and the constraints of our temporal sampling resolution, while also incorporating the reviewer's valuable suggestions for future research avenues.
- **(4) Strengthened narrative and integration**: Following the reviewer's suggesting, we have reorganized the Results and Discussion sections to improve logical flow, presenting findings from ecosystem-level 15N retention down to the underlying soil, microbial, and plant mechanisms. We have also strengthened the integration between sections, particularly in linking microbial processes to plant community outcomes.

We believe these comprehensive revisions have significantly enhanced the clarity, precision, and overall impact of our manuscript. We are grateful for the opportunity to improve our work and believe it now makes a stronger contribution to understanding N cycling in seasonally frozen grasslands.

**Specific Comments**

**1. Elongation of FTC:**

The manuscript assumes that climate change will lead to an elongation of FTCs. However, it is not clear why elongation rather than an earlier onset or other changes in FTC dynamics is expected, especially in temperate grasslands in China. We suggest the authors clarify this assumption and provide relevant references supporting the prediction of FTC elongation in their study region. This will strengthen the rationale for the study design and its climate relevance. We thank the reviewer for this important comment. We agree that clarifying why elongation of the freeze-thaw cycles (FTCs) period is a key projected outcome in our study region strengthens the climate relevance of our experimental design.

In the revised manuscript, we have amended the introduction (specifically in the climate context paragraph) to explicitly state the evidence for FTCs elongation and provide supporting references. The primary mechanism is asymmetric winter warming, which is particularly pronounced in northern temperate regions. This warming pattern does not simply shift the timing of a stable frozen period but extends the transitional seasons (autumn and spring) when soil temperatures fluctuate around

zero degree. This results in a later and less stable soil freeze-up in autumn and an earlier thaw in spring, thereby elongating the duration of the period susceptible to FTCs.

We believe this clarification, backed by the provided references, now solidly grounds our experimental treatment (intensifying FTCs within a potentially lengthened window) in a specific and credible climate change scenario for temperate grasslands.

**2. Hypothesis (1):**

The first hypothesis posits that intensified FTC would reduce retention of winter N resources due to physical disruption of soil aggregates, root damage impairing plant uptake, microbial cell lysis, and subsequent N leaching and denitrification losses. However, the study does not include direct measurements or assessments of the mentioned parameters, so it is not possible to robustly test this hypothesis. We recommend revising the hypotheses to focus on mechanisms and processes that are directly measured or can be reasonably inferred from the data.

We have rewritten our hypotheses to focus specifically on processes measurable with our 15N tracer data (i.e., contrasting plant community-level N retention, species-specific 15N acquisition), removing speculative mechanisms that were not directly measured (e.g., root damage, aggregate disruption). We have also ensured that the corresponding conclusions in the Results and Discussion sections are carefully aligned with this revised, measurable hypothesis. We thank the reviewer for this suggestion, which has significantly improved the clarity and scientific rigor of our work.

**3. Losses of winter N sources:**

The use of 15N tracers allows tracking of the fate of added labeled nitrogen, but it does not account for the dynamics of native, unlabeled N pools that may be mobilized during freeze-thaw cycles (e.g., through microbial lysis or mineralization of soil organic matter). This is particularly important given the observed increase in soil NH4+ concentrations, which likely reflects the release of native N rather than enhanced retention of applied 15N. The conclusion that intensified FTC did not lead to significant losses of winter N resources (e.g., abstract line 36; discussion lines 435, 452, 460–465) is therefore not fully supported by the data, as the study lacks direct measurements of N loss pathways such as leaching or gaseous emissions (e.g., NO3- leaching, denitrification, or volatilization). It would be valuable for the authors to clarify the scope and limitations of the 15N tracer method in assessing winter N retention, explicitly acknowledging that it only tracks added N and does not capture mobilization and potential loss of native soil N.

We sincerely thank the reviewer for this critical comment. In this revised MS, we have supplemented our data on leaching losses of  $^{15}$ N, which indeed show no significant increase (< 0.6%; Fig. 6) under intensified FTC treatments in the two grasslands. This new data provides direct experimental partly support for our

conclusion that significant losses of winter N sources via leaching did not occur.

The key improvements are follows:

- (1) **Incorporated new leaching data** (15N tracing data in deep soil, 30-50 cm): The leaching data has been added as a new figure and is described in the Results section. It provides direct evidence against significant hydrological N losses.
- (2) **Refined key conclusions**: The conclusion is now more precisely phrased as "did not lead to significant losses of the added winter 15N tracer", and we now explicitly cite the lack of elevated leaching losses as supporting evidence.
- (3) **Added a methodological limitation section**: We have added a dedicated paragraph in the Discussion 4.5 to explicitly acknowledge the scope and limitations of the 15N tracer method, as suggested by the reviewer.

We believe that the addition of the leaching data, coupled with the more nuanced interpretation of the 15N results, has substantially strengthened our manuscript and addresses the reviewer's concerns. We are deeply grateful for the insightful comment that led to this improvement.

**4. Plant-microbe interactions:**

The manuscript interprets the temporal pattern of stable microbial 15N retention and increasing plant 15N uptake as evidence of a decoupling mechanism that stabilizes winter N resources and fosters mutually beneficial plant-microbe interactions under intensified FTC. However, there is no observed decline in microbial 15N over time (Figure 6), which would be expected if immobilized N were later released to support plant uptake. Furthermore, the dynamic appears consistent across all treatments, suggesting it is a general feature of seasonal nitrogen cycling rather than a specific effect of FTC. Therefore, the conclusion that FTC induces a stabilizing mechanism through temporal decoupling is not directly supported by the data and should be removed.

We sincerely thank you for your critical and correct observation. Upon re-examination of our data, we fully agree with the opinion that the stable microbial 15N pool over time (Fig. 6) does not support the mechanism of a direct temporal transfer of N from microbes to plants, and that the original interpretation of a "mutually beneficial plant-microbe interaction" and specific "temporal decoupling" induced by FTC was an overstatement beyond what the data can support.

In this revised MS, we have rewritten the relevant sections in the Discussion (section 4.2) to refocus the narrative on the data-driven conclusions.

Our revised interpretation now emphasizes the following points, which are directly supported by our data:

(1) The soil pool acted as a major sink for the added 15N, with significantly elevated

recovery under high-frequency FTC (HFTC), suggesting efficient physical protection and chemical stabilization of the released N.

- (2) Microbes acted as a crucial biological buffer by rapidly immobilizing 15N in early spring, thereby securing the N pulse against potential loss during a period of low plant uptake.
- (3) The subsequent rise in plant 15N uptake is now discussed as likely originating from other soil N pools (e.g., the stabilized soil N or from ongoing mineralization), rather than from a direct release of the immobilized microbial 15N. We explicitly acknowledge that the stable microbial 15N pool argues against a direct transfer.

We have therefore reframed the ecosystem N retention mechanism as a combination of abiotic stabilization in the soil and initial biological immobilization by microbes, both of which are directly supported by our 15N data. We believe the revised text is now more accurate and robust, and we are grateful to the reviewer for their insight which has significantly improved the quality of our manuscript.

**5. Clarify scope of conclusion regarding FTC effects:**

The discussion (line 454) mentions that intensified FTC increased total 15N recovery, and the conclusion (line 538) states that intensified FTC reveals enhanced retention of winter N resources. However, the data show that this effect is limited to the high-frequency FTC treatment, with no comparable increase under low-frequency FTC. Therefore, the current phrasing could be misinterpreted as evidence of a general ecosystem response or resilience to FTC. Please clarify that the observed effect pertains specifically to high-frequency FTC.

We agree that our original phrasing was overly broad and could be misinterpreted as indicating a general response to any intensification of FTC. It is correct that the significant increase in total 15N recovery was indeed a specific effect of the HFTC treatment, as the low-frequency FTC (LFTC) did not elicit a comparable response.

We have carefully revised the manuscript to clarify the scope of this finding. Specifically:

- (1) **In the Discussion**, we have rephrased the statement to explicitly specify that the increase in total 15N recovery was driven by the high-frequency FTC treatment.
- (2) **In the Conclusions**, we have similarly amended the language to state that enhanced retention of winter N resources in spring was revealed specifically under intensified high-frequency FTC.

These changes ensure that our conclusions accurately reflect the specific conditions under which the observed effect occurred, preventing any potential misunderstanding about a general ecosystem resilience to FTC.

**6. Temporal resolution of sampling and N cycling processes:**

While the seasonal soil sampling intervals are appropriate for tracking broad patterns, the temporal resolution immediately before, during, and after the freeze-thaw treatments appears too coarse to capture short-term N cycling processes. Processes like nitrification and denitrification usually occur within days after FTCs and can contribute to substantial N losses in form of  $N_2O$  fluxes. Please consider acknowledging this limitation, especially when interpreting mechanistic effects of FTC on nitrogen transformations.

We appreciate you for raising this important point regarding the temporal resolution of our sampling. We acknowledge that our seasonal sampling intervals, while appropriate for tracking the broader seasonal patterns of plant N uptake, were likely too coarse to capture the rapid, short-term dynamics of microbial N cycling processes (such as immediate microbial 15N immobilization/remobilization or N2O fluxes) that occur within days following freeze-thaw cycles.

Our experimental design was primarily focused on investigating the fate of winter N sources in the context of plant N acquisition, which is a cumulative and ecologically decisive process over the growing season. From this perspective, our sampling strategy was sufficient to accurately quantify the ultimate utilization of the 15N tracer by plants.

We have followed your suggestion and explicitly acknowledge this limitation in the revised manuscript. A statement has been added to the Discussion (Section 4.5) to clarify that our interpretations of short-term mechanistic effects, particularly regarding microbial N transformations immediately following FTCs, are constrained by the temporal resolution of our sampling. We agree that future studies targeting these rapid microbial processes would benefit from higher-frequency sampling. We thank the reviewer for this valuable comment, which helps to better define the scope and interpretation of our findings.

**7. Soil sampling depth and deep roots:**

The study sampled only the top 20 cm of soil, which likely captures much of the microbial and plant root activity. However, nitrate is highly mobile and may leach below 20 cm, especially following FTC-induced mineralization, potentially leading to underestimation of N losses and overestimation of retention. Moreover, Hypothesis 2 suggests that deep-rooted species may increase winter N uptake under intensified FTC. Yet, root 15N retention was assessed only within the top 20 cm, which may not reflect uptake from deeper soil layers where such species may access nitrate. This could reduce the apparent contrast between shallow- and deep-rooted species. We recommend acknowledging these limitations when interpreting both N retention dynamics and species-level uptake patterns. We thank the reviewer for raising this important point regarding soil sampling depth.

We designed our soil sampling to a depth of 20 cm because approximately 80% of the root biomass is distributed within this upper soil layer in the studied grasslands. However, we agree that sampling only the top 20 cm could lead to an underestimation

of N leaching and provide an incomplete picture of N uptake by deep-rooted plant species.

Actually, our experimental design included deep soil sampling (30-50 cm). The data from this depth, now explicitly presented in Figure 6, show that the amount of  $^{15}$ N recovered below 20 cm was negligible (consistently < 0.6% of total recovery). This provides direct evidence that significant leaching of the added winter  $^{15}$ N tracer below the root zone did not occur, and that the top 20 cm effectively captured the vast majority of the retained tracer.

We have taken the following steps to address this limitation:

- (1) We have explicitly acknowledged this limitation in the revised Discussion 4.5. We state that our sampling to 20 cm, while capturing the majority of microbial activity and fine root biomass (> 80%) in the grasslands, may lead to an underestimation of N leaching below the root zone and may not fully reflect the N uptake by deep roots.
- (2) We have refined our interpretation of the species-specific results for Hypothesis 2. We now cautiously state that the measured 15N acquisition in the top 20 cm reflects the competitive outcome in the surface soil horizon. We clarify that while deep-rooted species might access resources from deeper layers, their enhanced 15N uptake in the surface soil still indicates a successful competitive strategy under intensified FTC. However, we concede that the absolute amount of N acquired by deep-rooted species might be higher than we recorded, and the contrast with shallow-rooted species might be even more pronounced in reality.

We believe that by acknowledging this limitation and refining our interpretations accordingly, we have provided a more accurate and nuanced discussion of our findings. We thank the reviewer for highlighting this important point.

**8. Species specifics (Hypothesis 2):**

You hypothesize that "intensified FTC would lead to differential utilization of winter N sources among plant species," and in the discussion, you refer to several plant traits to explain species-specific responses. However, it is currently difficult for the reader to follow these arguments without a clear overview of the relevant species and their functional traits. To strengthen the link between your hypothesis and interpretation, we recommend including a brief summary of the observed species and their key traits that you mentioned in your hypothesis 2 (competitive abilities, root system architecture (particularly rooting depth and winter root activity), temporal niche partitioning in growth phenology (early spring green-up), and susceptibility for root damage) in the Methods section. Further, we suggest to formulate the hypothesis more concretely. This would provide helpful context for understanding the mechanisms underlying your findings.

Thank you for this valuable suggestion. We agree that providing a clearer overview of the plant functional traits could strengthen the link between our hypothesis and interpretation. Accordingly, we have added Supplementary Table 1, which provides a concise summary of the key traits for the species studied, as recommended. Revised Hypothesis 2 to make it more specific, directly linking the expected outcomes to the functional traits described.

**9. Site specifics:**

The manuscript provides useful site-specific information; however, the methods section does not specify how these data were obtained—whether from field measurements, previous studies, databases, or modeling (e.g. bulk density). Including details on the sources and collection methods for these site parameters is important for transparency and reproducibility. Moreover, while site differences are documented, the manuscript lacks a discussion of how these environmental and edaphic differences may have influenced the observed results. Since the study compares two distinct sites, integrating an analysis or interpretation of how site characteristics might drive differences in nitrogen cycling, microbial activity, plant uptake, or freeze-thaw responses would strengthen the ecological context.

Thank you for these insightful and constructive suggestions. We have addressed these points comprehensively in the revised MS:

**(1) Enhanced methodological transparency:**

In the Methods section (now Section 2.1), we have specified the source and collection method for each key site parameter. We explicitly state that soil bulk density, texture, pH, total C and N were determined from our own field measurements and laboratory analysis of soil samples collected during the study establishment. Meteorological data were obtained from the China Meteorological Administration, with specific station codes and access details now provided.

**(2) Integrated site comparison in Discussion:**

As suggested, we have added a dedicated analysis in the Discussion (Section 4.3) that explicitly links the observed responses between the two grasslands to their contrasting environmental and edaphic conditions.

Initially, we hypothesize that at an ecosystem level, the effects of intensified FTC on retention of winter N resources would be mediated by ecosystem type, with the sandy steppe experiencing a greater reduction than the meadow steppe due to its inferior edaphic conditions (Table 1). Contrary to our first hypothesis, plant 15N recovery showed no significant difference between the meadow and sandy steppes under intensified FTC.

In the revised MS, the text as follows:

"4.4 Ecosystem-Level Convergence in Plant N Uptake Despite Divergent Soil

**Conditions**

At the ecosystem level, the absence of a significant difference in the magnitude of plant 15N uptake between the meadow steppe and sandy steppe is noteworthy, given their pronounced differences in soil fertility (Table 1). This apparent convergence can be explained by several compensatory mechanisms that operated across the two contrasting sites.

First, the dynamics of nitrogen availability differed. While the meadow steppe exhibited a higher net N mineralization rate in early spring, providing a larger initial pulse of inorganic N, the sandy steppe likely compensated through more efficient plant uptake of the available NH4+-N pool, as indicated by its significantly lower soil NH4+-N concentrations. This suggests that plant communities in the resource-limited sandy steppe are adapted for rapid nitrogen acquisition when it becomes available.

Second, the physical pathway of nitrogen loss was similarly constrained in both ecosystems. The lack of a significant difference in 15N leaching losses indicates that intensified FTCs did not disproportionately enhance hydrological N losses in the coarser-textured sandy steppe soil. This physical retention created a similar baseline condition for nitrogen conservation in both systems.

Finally, the biotic component in the sandy steppe demonstrated inherent resilience. The microbial community, adapted to arid and nutrient-poor conditions, likely possessed metabolic traits that buffered it against FTC-induced physiological stress. This microbial stability, coupled with a plant community dominated by deep-rooted and drought-tolerant species, contributed to an ecosystem-level N retention capacity that was functionally comparable to that of the more nutrient-rich meadow steppe. Therefore, the similar levels of plant 15N uptake emerged not from identical processes, but from different yet effective strategies in each ecosystem."

We believe these revisions significantly improve the transparency, reproducibility, and ecological depth of our study.

**10. Meteorological information:**

The manuscript references the China Meteorological Administration website (http://data.cma.cn/) as the source for meteorological information at the study sites. However, this website is primarily in Chinese and can be difficult to navigate for non-Chinese speaking readers. To improve accessibility and reproducibility, we recommend providing a more direct and specific link to the exact data pages used, or alternatively, suggesting an English version or database where the meteorological data can be accessed more easily by an international audience. This would help readers verify the data and facilitate broader use of the study's findings.

Thank you for this valuable suggestion to improve the accessibility and reproducibility of our meteorological data. We have added another international

website: NOAA Integrated Surface Database (ISD):

(https://www.ncei.noaa.gov/maps/hourly/) in the revised MS, as an alternative or for verification. This database integrates global surface meteorological observations and includes data from numerous Chinese international exchange stations.

**11. Artificial manipulation and microclimate measurements:**

The study aims to address the lack of natural in situ FTC experiments (lines 71), but the use of polyester tents and air heating still introduces some artificial influences. While the manuscript notes that mesh windows were used to reduce CO2 accumulation, it would still be helpful to clarify how other potential microclimatic changes were accounted for. For instance, air temperature was recorded at 5 cm above ground (line 174), but these data are not presented or discussed. Changes in humidity, photosynthetically active radiation (PAR), or CO2 levels could still affect plant growth and nitrogen uptake. We recommend briefly discussing these possible side effects of the experimental setup to help contextualize the findings. Also, figure 2 shows that the soil processing period overlaps with the occurrence of freeze-thaw cycles at the sandy steppe site. Please address if this overlap may have influenced the results.

We thank the reviewer for raising these important methodological considerations. We have addressed each point as follows:

- (1) **Contextualizing our in-situ approach**: While we acknowledge that the use of tents introduces some artificiality, our experimental design represents a significant improvement over previous laboratory-based FTC simulations. Unlike studies that transport soils to the lab for temperature manipulation, our in-situ approach maintains natural soil structure, root networks, and microbial communities, thereby providing a more ecologically realistic representation of FTC impacts in intact grassland ecosystems.
- (2) **Temperature monitoring clarification**: We apologize for the unclear description in our original manuscript. We have now clarified that while soil temperature at 10 cm depth was continuously monitored, the 5 cm depth was periodically verified with a handheld thermometer specifically during FTC treatments to ensure target temperature thresholds were met. The continuous 10 cm depth data, which effectively captures the FTC dynamics. We have corrected this methodological description in the revised MS to avoid any confusion.
- (3) **Microclimate considerations**: We acknowledge that we did not monitor humidity, PAR, or precise CO2 levels due to equipment limitations. We recognize that measuring these parameters would have provided a more complete picture of the microclimatic changes induced by our tents. We also state that this insightful comment highlights an important aspect for future research, and we will incorporate the monitoring of these key microclimatic parameters in subsequent experiments to provide a more comprehensive understanding in the future.

However, we note that: (i) our treatments were applied before plant green-up when vegetation would be less responsive to subtle microclimatic variations; (ii) the mesh windows substantially reduced CO2 accumulation; and (iii) the short duration of tent deployment (6-12 cycles, immediately removed after treatment) minimized potential impacts on plant growth and N uptake. We agree that monitoring these parameters in future studies would provide valuable supplementary data.

(4) **Treatment timing**: We confirm that our experimental FTC treatments were completed before the natural FTC period began, as clearly shown in Figure 2a,b. The treatments were intentionally scheduled approximately 15 days prior to the natural FTC period to simulate FTC elongation while avoiding overlap with natural cycles, thus preventing confounding effects.

We believe these clarifications and methodological improvements address the reviewer's concerns while demonstrating the ecological validity of our experimental approach.

**12. Soil moisture measurement and implications of differences:**

While soil moisture data were recorded using data loggers (line 205), the manuscript does not specify the sensor types, calibration methods, or how values (in m³ m⁻³) were derived. The presence of negative soil moisture values suggests possible measurement or calculation errors that should be addressed. Additionally, there is an inconsistency between the text (lines 280–284), which states that elevated soil moisture occurred only in early spring, and Figure 2b, which appears to show sustained increases under LFTC and HFTC throughout much of the season. It should be included in the discussion how treatment-induced changes in soil moisture may have influenced nitrogen dynamics and plant ¹⁵N uptake, as it was a significant predictor in the correlation and random forest analyses.

We apologize for the absence of these essential methodological details. We have now revised the Methods section to include the following information:

Soil volumetric water content (VWC, m³/m³) and temperature were monitored using a HOBO H21-002 data logger (Onset Computer Corporation, USA) coupled with 10HS soil moisture sensors. The 10HS sensor estimates VWC by measuring the soil dielectric permittivity at a frequency of 70 MHz. The sensors were deployed with their factory-predefined standard calibration equation, which directly converts the measured dielectric readings into volumetric water content values (m³/m³). Therefore, for the conventional soils in our study, the data logger directly outputs the final VWC values, and no further calculations were required by us. The negative values occurred primarily in cold and frozen soil conditions and are a known artifact of the sensor's calibration at the extremely lower end of its measurement range. In our revised dataset, all negative VWC values have been set to 0 m³/m³, reflecting that the liquid water content was at or below the sensor's effective detection limit. This is a standard

data-cleaning practice, and we have added a note in the methods section to state this correction. The number and magnitude of these values were negligible and did not influence the statistical outcomes or overall conclusions.

Upon re-examination, we confirm that Figure 2b is accurate: the LFTC and HFTC treatments led to a sustained increase in soil moisture over much of the seasons. We have therefore corrected the text in the Results section to accurately reflect the figure.

We fully agree that exploring the mechanistic link is crucial. We have now added a dedicated paragraph in the Discussion section to elaborate on this. In this new paragraph, we explicitly state that soil moisture was a key predictor in our statistical models, discuss how the treatment-induced increases in soil moisture could have created conditions that enhanced microbial activity and nitrogen mineralization.

**13. Restructure Chapter 2.3 Sampling and Processing:**

This section would benefit from restructuring to more clearly distinguish which subsamples were used for which analyses and to detail the analytical procedures more consistently. We would recommend to reorganizing this section to clearly present the workflows for each measured variable (e.g., soil mineral N, DOC, microbial C and N, soil/plant/microbial 15N, soil moisture and temperature) to improve readability and reproducibility.

Further, it is unclear whether the described K2SO4 extraction and analysis refer only to microbial biomass C and N samples or if the same procedure was used for soil mineral N (NH4+ and NO3-) and DOC analyses as well (lines 231-233). Please clarify whether different extraction procedures were used for mineral N and DOC, and if so, provide the details (e.g., solution type, shaking duration, soil-to-solution ratio).

Regarding the 15N measurements, plant and soil 15N were measured with an elemental analyzer coupled to IRMS — was the same system used for total microbial 15N, or was a different method used? Please also clarify whether the same elemental analyzer (Elementar Vario Max CN) was used for all C/N and 15N analyses, or if multiple instruments were involved.

- (1) We have thoroughly restructured and revised Chapter 2.3 to present a clear, workflow-based structure according to your recommendations. The revised section now contains dedicated sub-sections for each major group of analyses, as follows:
- 2.3.1. Soil moisture and temperature
- 2.3.2. Soil and plant properties
- 2.3.3. Soil Microbial Biomass
- 2.3.4 15N level in soil, plant and microbe

This new structure explicitly outlines which subsamples were used for each analysis and details the analytical procedures in a logical sequence, significantly enhancing

readability and reproducibility.

(2) Different extraction procedures were indeed used for different analyses, and we have now provided distinct details for each in their respective sub-sections.

**Soil inorganic N and DOC**: fresh soil samples were extracted with 2 M KCl at a 1:5 soil-to-solution ratio (10 g fresh soil with 50 mL KCl) by shaking for 1 hour on a mechanical shaker. The extract was then filtered and used for the determination of NH4+ and NO3-, as well as for DOC analysis. All plant and soil elemental (C/N) were performed using elemental analyzer (Elementar analyzer Vario MAX CN, Germany).

**Microbial biomass C and N**: the chloroform fumigation-extraction method was employed. Both fumigated and non-fumigated soils were extracted with 0.5 M K2SO4 at a 1:4 soil-to-solution ratio (15 g fresh soil with 60 mL K2SO4) by shaking for 30 minutes and then filtered.

15N analyses: 15N analyses for plant and soil samples were performed using elemental analyzer (Elementar analyzer Vario MAX CN, Germany) coupled in continuous flow mode to a Isoprime Precision isotope ratio mass spectrometer (IRMS) (Isoprime, USA). 15N analyses of microbial biomass was determined using an modified diffusion method (with slight heating and acid-soaked glass fiber filters as the trap), and the filters containing the absorbed N were then measured using the same EA-IRMS system.

**14. Correlation analysis and random forest:**

The manuscript includes correlation and random forest (RF) analyses to identify key predictors of plant 15N acquisition, incorporating variables such as DOC, microbial carbon, and microbial community composition (bacterial vs. fungal biomass). However, the rationale for including both statistical approaches is not clearly explained, and the results from these analyses are not sufficiently integrated into the discussion. It remains unclear how the outputs of both approaches complement each other, and what ecological insights they offer regarding nitrogen dynamics under freeze-thaw conditions. Additionally, the manuscript does not explain the relevance of DOC and microbial biomass C to nitrogen cycling or FTC effects. Similarly, the ecological importance of differentiating microbial groups (bacteria vs. fungi) is not introduced and not interpreted in the results or discussion. Lastly, it is unclear why these analyses were performed only for the two FTC treatments and not for the control. To improve coherence, we recommend explaining the rationale for including both correlation and RF approaches and discussing the ecological relevance of the identified predictors.

We thank the reviewer for these insightful comments regarding our statistical approaches. We have partly revised the relevant sections to address these concerns as follows:

**1. Clarified rationale for using both correlation and random forest analyses:**

In the revised Methods section (Statistical Analysis), we now explicitly explain the complementary purposes of these two approaches:

- (1) Correlation analysis was used as an initial screening tool to identify potential relationships between environmental factors and plant 15N acquisition across different treatments.
- (2) Random Forest analysis was then employed as a more robust machine learning method that can handle high-dimensional data and minimize overfitting, while effectively ranking variable importance and handling collinearity among predictors.

**2. Enhanced ecological interpretation of key predictors**

We have significantly expanded the Discussion to provide proper ecological context for the identified predictors:

- (1) The relevance of DOC and microbial biomass C to N cycling under FTC conditions is now explicitly discussed, particularly their roles in microbial metabolism and N immobilization processes.
- (2) We have removed soil bacterial and fungal biomass from the revised MS, as these variables were found to be less important predictors in our Random Forest analysis and did not contribute significantly to explaining plant 15N acquisition patterns.

**3. Separate analyses for all treatments**

Following the reviewer's suggestion, we have now performed and presented separate correlation and Random Forest analyses for the Control, LFTC, and HFTC treatments in the revised Figures 7 and 8. This approach allows for clearer comparison of how key drivers of plant 15N acquisition change across different treatment intensities.

**4. Improved integration of statistical results**

The Results section now more clearly presents the outputs of both analyses, while the Discussion section provides a synthesized interpretation of what these statistical approaches collectively reveal about the mechanisms controlling plant N acquisition under FTC conditions.

We believe these revisions have significantly improved the coherence and ecological insight derived from our statistical analyses, and we thank the reviewer for these valuable suggestions.

15. Discussion on the relevance and magnitude of observed differences: While the manuscript highlights statistically significant differences between treatments, it does not sufficiently address the ecological or functional relevance of these differences. A more in-depth discussion is needed on the magnitude of the changes observed. Please consider discussing whether the observed changes are likely to have substantial impacts on grassland resilience, nutrient cycling, or plant community structure in the context of winter climate change.

We thank the reviewer for this important comment regarding the ecological relevance of our findings. We have substantially revised the Discussion to provide a more in-depth analysis of the potential ecological consequences of the observed changes, focusing specifically on their implications for grassland resilience and community structure.

In the revised MS (Section 4.3), we now explicitly discuss the functional significance of the species-specific shifts in N acquisition. While the absolute magnitude of changes in 15N uptake for individual species might appear limited in the short term, we argue that their "directional consistency" is ecologically meaningful. The amplified competitive advantage for cold-adapted species like *S. baicalensis* and *H. mongolicum*, coupled with the simultaneous suppression of subordinate species, represents a redistribution of N resources that could alter plant community composition if sustained over multiple years. This is particularly relevant in the context of winter climate change, as repeated FTC events may cumulatively favor stress-tolerant species, thereby reducing functional diversity and potentially shifting the community toward a new state.

Furthermore, we link these plant-level responses to ecosystem resilience. The overall reduction in community-level 15N acquisition under high-frequency FTC, despite the stability of the soil and microbial N pools, suggests a potential decoupling between ecosystem N retention and plant N utilization. This indicates that resilience, defined as the capacity to maintain both structure and function, may be challenged, as the ecosystem's ability to conserve N does not directly translate to unchanged plant resource acquisition.

By framing our results in terms of these longer-term, cumulative ecological processes, we have strengthened the discussion of the functional relevance of our findings beyond immediate statistical significance.

**16. Future research directions and limitations of the study:**

The future research section outlines useful directions, particularly regarding microbial functional traits and long-term 15N fate. However, other potentially impactful avenues are overlooked. For instance, it would be valuable to differentiate between the origins of newly available nitrogen during freeze-thaw cycles, specifically, whether it stems from microbial cell lysis, root mortality, or physical disruption of soil aggregates. Distinguishing these sources could significantly enhance mechanistic understanding of nitrogen retention and loss pathways. Additionally, the study briefly references nitrogen losses but does not address gaseous emissions. Monitoring greenhouse gases (e.g., N2O, CO2) during FTC events could offer further insight, especially given that FTC induced N2O peaks often occur without corresponding CO2 increases, which could have a relation to your results e.g. on increases in microbial biomass N and decreases in microbial biomass C.

We have completely rewritten and expanded the "Limitations and future work" section (now Section 4.4) to incorporate these specific suggestion.

Our revision as follows:

**Limitations and future work.**

**(1) Methodological constraints:**

First, while our 15N tracer approach precisely tracked the fate of winter inorganic N, it cannot account for the dynamics of the native soil N pool, including mobilization and loss pathways of unlabeled N. Second, the temporal resolution of our sampling, while appropriate for quantifying seasonal patterns of plant N uptake, was insufficient to capture rapid microbial N transformations and gaseous fluxes occurring within days following FTC events. Third, our 20 cm soil sampling depth, though capturing the majority of root activity in these grasslands, may not fully reflect N dynamics in deeper layers where deep-rooted species access resources and nitrate leaching could occur.

- (2) Experimental design considerations include potential microclimatic effects of our tent-based FTC manipulation. While we employed mesh windows to minimize CO2 accumulation and implemented short-duration treatments, we did not monitor humidity, photosynthetically active radiation, or precise CO2 levels, which could have provided additional context for interpreting plant responses.
- (3) Contextual limitations should be noted. Our findings from two contrasting temperate grassland types provide important insights but may not be directly transferable to other ecosystems with different soil properties, microbial communities, or plant functional compositions. Additionally, the single-year duration of our experiment limits our ability to assess long-term ecosystem adaptations to repeated winter climate perturbations.
- **(4) Differentiation of N sources.** We now highlight the importance of distinguishing the specific origins of newly available N, whether derived from microbial cell lysis, root mortality, or physical disruption of soil aggregates, as a key priority for future research. We acknowledge that such differentiation would substantially enhance our mechanistic understanding of nitrogen retention and loss pathways during FTC events.
- (5) Gaseous emission monitoring. We have incorporated the reviewer's valuable suggestion regarding simultaneous monitoring of greenhouse gases (particularly N2O and CO2) with high temporal resolution during FTC events. We specifically note that this approach would help elucidate the relationships between microbial C and N cycling, especially relevant given the decoupled responses of microbial biomass C and N observed in our study.

These limitations, however, clearly define valuable avenues for future research, including high-resolution tracking of coupled C-N gas fluxes, molecular characterization of microbial functional traits, differentiation of N sources from various soil pools, and long-term 15N tracing across multiple freeze-thaw seasons. We believe this comprehensive revision significantly strengthens the forward-looking

impact of our manuscript, and we are grateful for the reviewer's guidance in helping us identify these key research gaps.

**Technical Corrections**

1. Line 273-395 Results: We suggest removing redundant "p < 0.05" and instead define significance threshold in the Methods or where relevant, report actual p-values.

We have removed redundant "p < 0.05" and defined significance threshold in the Methods.

2. Figure 2: We recommend to make the figure broader to better see the single FTCs. The labeling of the dates is unclear. The legend for samplings looks like another sampling event which is confusing.

Revised as suggested. We have made the figure broader, added the label of dates and changed the legend for samplings to better show our data.

- 3. Figure 3, 4 and 6: The dots for single measurements are not necessary and just confusing if they overlap having error bar should be enough. Revised as suggested. Please Figure 3, 4 and 6.
- 4. Figure 5 and 7: Both are exactly the same plot, figure 5 doesn't fit to the description the figure showing plant biomass N is missing.
  We apologized for our mistake. We have upload a new Figure 5 in the revised MS.
- **5.** Figure 3 to 7: please including the exact dates of sampling in the legend. Revised as suggested. We added the exact dates of sampling in the legend. Please see Figure 3 to 7. And we have indicated the sampling time in Section 2.3 Sampling and Processing.
- 6. Introduction line 37: replace "while" with "While" Revised as suggested.
- 7. Line 106: does -2-1°C mean -2.1 °C? It means -2 °C to 1 °C.
- 8. Line 125: replace "the predominant soil type in meadow grassland is loam soil, and which in sandy grassland is sandy loam soil." With "the predominant soil type in meadow grassland is loam, while in sandy grassland it is sandy loam."

Revised as suggested.

9. Line 131: add space "78 %" Revised as suggested.

10. Line 161: There is a verb missing: ", and no significant differences in plant/microbial N concentrations when compared to the 15N treatments."

Revised as suggested.

11. Line 172: add space "15 cm"

Revised as suggested.

12. Line 231: add space "60 ml"

Revised as suggested.

13. Line 232: How much Molar was the K2SO4 solution?

"0.5 Molar", we have added this.

14. Line 232: add space "30 min"

Revised as suggested.

15. Line 234: please state for what the conversion coefficient is used.

The conversion coefficient is 0.45.

16. Line 244: replace "as well as microbial community structure in situ soils" with "as well as the microbial community structure of in situ soils" Revised as suggested.

17. Line 250: The part with the calculation of 15N acquisition/recovery is not statistical analysis.

Thanks for pointing out. We have changed this section to 2.3 Sampling and Processing.

18. Line 253: replace ";" with ","

Revised as suggested.

19. Line 258: add space "x V x"

Revised as suggested.

20. Line 259: start new paragraph

Revised as suggested.

21. Line 264: Is "rcorr" from R or SPSS?

"Rcorr" is from R.

22. Line 266: randomForest and rfPermute packages from R or SPSS?

"RandomForest" and "rfPermute" packages are from R.

**23. Line 271: remove comma "SigmaPlot 14.0"**

Revised as suggested.

**24. Line 271: Origin 14.0 is not existing**

Thanks for pointing out. We have changed this to "Origin 2021".

25. Line 335: replace "5N" with "15N"

Revised as suggested.

26. Line 377: add space "C and" and replace "Under" with "under" Revised as suggested.

27. Line 388: replace "plants" with "plant"

Revised as suggested.

28. Line 400: subsequent growing seasons would mean several years as there is one growing season per year

Thank you for this suggestion. The text has been modified to improve clarity.

29. Line 414: Abbreviation of MBN was already introduced

Revised as suggested.

30. Line 421: remove "."

Revised as suggested.

31. Line 436: specify that you mean plant water uptake

Revised as suggested.

32. Line 457: A verb is missing: "indicating that effective ecosystem-level N retention mechanisms."

Revised as suggested.

---

## Author Response (AR2)

**Notification to the authors 1 (03 Dec 2025)**:

Please do not separate the figure captions from the actual figures. Please directly place the individual figure captions below the respective figure image.

Thanks for your reminder. We have placed the individual figure captions below the respective figure image.

**Notification to the authors (01 Jul 2025)**:

Figure 1 may contain a territory that is disputed according to the United Nations. If and when the manuscript is accepted for final revised publication, you will be asked to choose one of the following options: (a) you could remove the disputed territory from the map and submit new figure files, or (b) we could add a statement that some figures contain disputed territories.

Thanks for your reminder. We potentially chose (b) to add a statement that Figure 1 may contain disputed territories, but we are unclear in which section of the manuscript it would be more appropriate to place the statement. It is important to note that our map is based on and solely depict the territory of China (Please see http://bzdt.ch.mnr.gov.cn/).

We are looking forward to your guidance.

On behalf of all authors, best regards,

Linna Ma